# Complexity of modular neuromuscular control increases and variability decreases during human locomotor development

Francesca Sylos-Labini [1,2✉], Valentina La Scaleia[2], Germana Cappellini[1,2], Arthur Dewolf [1], Adele Fabiano[3], Irina A. Solopova[4], Vito Mondì [5], Yury Ivanenko [2] & Francesco Lacquaniti [1,2✉]

When does modular control of locomotion emerge during human development? One view is that modularity is not innate, being learnt over several months of experience. Alternatively, the basic motor modules are present at birth, but are subsequently reconfigured due to changing brain-body-environment interactions. One problem in identifying modular structures in stepping infants is the presence of noise. Here, using both simulated and experimental muscle activity data from stepping neonates, infants, preschoolers, and adults, we dissect the influence of noise, and identify modular structures in all individuals, including neonates. Complexity of modularity increases from the neonatal stage to adulthood at multiple levels of the motor infrastructure, from the intrinsic rhythmicity measured at the level of individual muscles activities, to the level of muscle synergies and of bilateral intermuscular network connectivity. Low complexity and high variability of neuromuscular signals attest neonatal immaturity, but they also involve potential benefits for learning locomotor tasks.

[1] Department of Systems Medicine and Center of Space Biomedicine, University of Rome Tor Vergata, 00133 Rome, Italy. [2] Laboratory of Neuromotor Physiology, Istituto di Ricovero e Cura a Carattere Scientifico Fondazione Santa Lucia, 00179 Rome, Italy. [3] Neonatology and Neonatal Intensive Care Unit, Ospedale San Giovanni, 00184 Rome, Italy. [4] Laboratory of Neurobiology of Motor Control, Institute for Information Transmission Problems, 127994 Moscow, Russia. [5] Neonatology and Neonatal Intensive Care Unit, Casilino Hospital, 00169 Rome, Italy. ✉email: francesca.sylos.labini@uniroma2.it; lacquaniti@med.uniroma2.it

Two related concepts have attracted considerable interest in neuroscience and evolutionary developmental biology, modularity, and complexity[1,2]. Modularity refers to the existence of decomposable processes with stereotypical functions, interactions between components being weak compared with those within components[3]. Complexity refers to the minimum number of components (dimensionality) needed to account for a given behavior across a number of conditions[4]. Both low-dimensional and high-dimensional modular structures bring their own benefits, the former entailing better generalization across different contexts and the latter better separability of encoded information[5].

These issues are relevant also to the field of motor control[6–8]. The application of methods such as unsupervised machine learning algorithms to electromyographic (EMG) activities recovers the statistical structure of neural drive to the muscles from the time profiles of muscle activities. Thus, it has been shown in several different animal species that muscle coordination results from the modular engagement of groups of muscles (muscle synergies) sharing common temporal patterns of activation[7–9]. The temporal activation patterns and the muscle synergies reflect the output of spinal networks of premotor interneurons[10,11]. Modular synergistic organization may extend up to the level of motor cortex[12] and down to the level of subsets of individual motoneurons[13]. Dimensionality of control (i.e., the number of neuromuscular modules) has been addressed for several motor behaviors[14,15], and in particular for locomotion[16–20].

While the modularity of mature neuromuscular control of locomotion is relatively well established, its ontogenetic basis is still hotly debated for different animal species, especially altricial species such as rodents and primates[21–24]. Although these animals show a rich repertoire of primitive movements (such as general movements, kicking, or stepping[25]) well before birth which persist over a variable time after birth, the structure of the motor commands underlying these movements is still under dispute. Moreover, how innate and learned factors contribute to the progressive shaping of motor commands remains poorly understood. In this regards, different hypotheses have been proposed[21–27]. According to the learning hypothesis, modularity is not innate; infants and pups begin with a proliferation of variable, unstructured movement patterns, and experience teaches them to select coordinative solutions tailored to the current relations between their body and the environment. By contrast, a strict nativist view holds that locomotor modules are determined in early development, and are then robustly conserved into adulthood. A third hypothesis, more in line with current views in developmental neuroscience, is that both innate and learnt components play a critical role in motor development. In particular, one possibility is that the set of primitive modules revealed in stepping neonates is incomplete, and new fundamental modules are added during development to integrate postural and locomotor control.

The learning hypothesis is supported by the observation that infant stepping is highly variable, depending on subtle changes in context and neural or non-neural factors. Stable solutions would emerge only after several months or years of exploration of alternative, temporary solutions[26,27]. Consistent with this hypothesis, computer simulations emphasizing rich (high-dimensional) models with a weakly-defined prior structure are capable of learning locomotor tasks by generating efficient control policies based on generic neural networks[28].

On the other hand, the strict nativist view is supported by the observation that the isolated spinal cord of neonatal rodents stimulated by different neurotransmitters can express all patterns of EMG[29] and motoneuron[30] activity typical of mature locomotor coordination. Also, the muscle synergies of rats who underwent spinal transection as adults differ insignificantly from those of neonatal injured rats, despite different developmental histories[31].

Finally, the hypothesis of incompleteness of the neonatal locomotor modules is supported by the observation that the number of temporal patterns of muscle activation is low at birth, and reaches adult-like conformation only after independent, unsupported walking is established[24,32–37]. On the computational side, low-dimensional, flexible motor primitives encoded in spinal modules succeed in learning high-dimensional locomotor control problems of humanoids using reinforcement learning algorithms[38].

The hypothesis that components of early-established motor circuits are substrative and retained in adults, but their connectivity undergoes major reconfiguration during maturation[34] is also supported by work on animal models[24,39]. Thus, while spinal motor circuits are established early during development allowing animals to generate a variety of movements prior to and at birth[25,40], the activity of these motor circuits changes drastically during development[41].

The different hypotheses outlined above may be resolved experimentally by carefully comparing the EMG activities at several different stages of motor development. In particular, measures of EMG complexity and variability may contribute to the resolution. The identification of neuromuscular modules in infants is made difficult by the presence of considerable noise, high intra-individual (inter-step), and inter-individual variability in EMG activity[32,42,43]. Also, the criteria to determine the number of basic modules (the dimensionality issue) vary in different studies, which in turn may influence the interpretation of the spatiotemporal muscle activity. For instance, using a fixed threshold for the variance accounted for (VAF) by the selected modules may not be reliable when variable amounts of noise affect the overall output. In fact, a systematic method to determine the number of the basis vectors has not been established by now[44].

Here, we address the issue of modularity and complexity of neuromuscular control of stepping during human development using multiple, complementary approaches. To this end, we considered cross-sectional data spanning distinct developmental stages, from birth till adulthood, with several different time points during the first 3 years of age. To provide context, we also analyzed artificial data sets with varying amounts of simulated noise that matched the real data but had known, preset dimensionality. We used three different methods for EMG factorization, spatial decomposition, temporal decomposition, and space-by-time decomposition[45]. Each of these methods makes specific assumptions: the spatial decomposition assumes spatial modularity, the temporal decomposition assumes temporal modularity, and the space-by-time decomposition assumes the concurrent existence of spatial and temporal modules[37,45]. In addition, we assessed dimensionality from a frequency analysis of EMG activities of individual muscles including both periodic and aperiodic components[46], and from bilateral EMG network analysis[47].

Dimensionality was quantified by means of both VAF and consistency measures well suited to take the presence of noise into account. Consistency was estimated as a similarity index of the temporal patterns across all strides for the spatial decomposition, a similarity index of the synergies across all strides for the temporal decomposition, and as a diagonality index of the activation coefficients across all strides for the space-by-time decomposition. We found that the rate of change of these consistency measures as a function of the hypothetical number of modules was a sensitive indicator of the dimensionality of the modules in the presence of noise.

In this manner, we provide a global assessment of modular complexity during locomotor development by reporting a systematic trend toward the increase of dimensionality from the neonatal stage to adulthood across all employed measures. Overall, the results are compatible with the hypothesis that infants begin with a small set of primitive neuromuscular modules, and add more modules to better integrate sensory and cortical signals in locomotor control. We also show that the increase of complexity from the neonatal stage to adulthood involves multiple levels of the motor infrastructure, from the intrinsic rhythmicity measured at the level of individual muscle activities, to the level of muscle synergies and bilateral intermuscular network connectivity. We find that, while the complexity of neuromuscular organization increases with age, in parallel the associated variability decreases.

## Results

**General gait parameters**. Table 1 summarizes the characteristics of the participants in each group, the average stride duration, and its variability for each group. As reported in previous studies[33,43], stride cycle duration T significantly shortened with increasing age, reaching an average duration in adulthood that was less than half of the duration at birth. On the other hand, its variability was more than 20 times larger in neonates compared to adults, and systematically decreased with age in children.

Except as otherwise remarked, all EMG analyses were performed on data concatenated over 7 strides for each muscle. In order to have a comparable set of data for all participants, the EMG data of 7 strides at the same (treadmill) or similar (on ground) speed were used for all analyses. The constraint of 7 strides arose from the results obtained in neonates, who typically perform a limited number of steps[27,32,36,37]. Here, we found that 7 strides was the maximum number of strides that was common to all our neonates. In all participants, we recorded the EMG activity of rectus femoris (RF), biceps femoris (BF), tibialis anterior (TA), and gastrocnemius lateralis (LG) of both lower limbs.

Figure 1 shows results from two representative participants, a neonate (a) and an adult (b). Despite higher variability (in stride duration and EMG amplitude) in the neonate, one can notice a tendency for synchronous activation of groups of muscles (BF, RF, LG), denoting a lower complexity of neuromuscular activations compared to adults. Notice that these same muscles are asynchronously activated in the adult. In the next sections, we will consider the two features, complexity and variability of muscle coordination, in more detail.

**Intra-muscular complexity**. We first consider intra-muscular complexity, as revealed by the frequency contents of the EMGs of individual muscles in neonates and adults. We applied an algorithm[46] that modeled the power spectral density (PSD) of the rectified and time-interpolated EMG data as a combination of periodic and aperiodic components (Fig. 2a–d). Model fitting was accurate, with average $r^2 = 0.82$ (95% confidence interval $CI_{95\%}$, [0.81, 0.83]).

For each muscle, the periodic component of PSD had a significantly greater number of peaks in adults than in neonates (Fig. 2b, Wilcoxon rank sum test, $p < 0.001$ for BF, $p = 0.011$ for RF, $p = 0.033$ for LG, and $p = 0.0036$ for TA). Moreover, in adults the center frequency of the peaks was more consistent than in neonates (Fig. 2c). Indeed, the great majority (~83%) of the peaks of the periodic component of the EMGs of adults was located near the stride frequency (1/T) and its first four harmonics (from 2/T to 5/T). By contrast, in neonates, ~17% and ~10% of the peaks were located around 1/T and 2/T respectively, and the

**Table 1 Characteristics of subjects and general stepping parameters.**

| Group | | N | Sex F | Sex M | Age | Weight, kg | Limb length, cm | Average stride duration, s | Stride duration variability (SD), s |
|---|---|---|---|---|---|---|---|---|---|
| Neonates | | 11 | 7 | 4 | 6.4 (±4.9) | 3.2 (±0.3) | 19 (±1) | 2.67 (±0.46) | 0.44 (±0.16) |
| Infants | g1 | 7 | 4 | 3 | 4.8 (±0.7) | 6.3 (±0.2) | 22 (±1) | 2.25 (±0.62) | 0.53 (±0.25) |
| | g2 | 7 | 2 | 5 | 7.3 (±0.3) | 8.3 (±0.7) | 25 (±1) | 1.68 (±0.33) | 0.36 (±0.18) |
| | g3 | 5 | 1 | 4 | 9.7 (±0.2) | 9.4 (±1.5) | 26 (±2) | 1.48 (±0.38) | 0.25 (±0.14) |
| | g4 | 11 | 3 | 8 | 11.8 (±1.3) | 10.6 (±1.7) | 27 (±1) | 1.33 (±0.45) | 0.22 (±0.13) |
| Toddlers | | 15 | 8 | 7 | 12.9 (±1.1) | 9.6 (±0.7) | 27 (±2) | 1.05 (±0.24) | 0.17 (±0.12) |
| Preschoolers | | 8 | 3 | 5 | 37.4 (±8.4) | 13.2 (±3.3) | 38 (±4) | 0.94 (±0.15) | 0.15 (±0.03) |
| Adults | | 15 | 7 | 8 | 27.5 (±9) | 68.5 (±9.7) | 77 (±7) | 1.15 (±0.08) | 0.02 (±0.01) |

Age is in days for neonates, months for infants, toddlers and preschoolers, and years for adults.

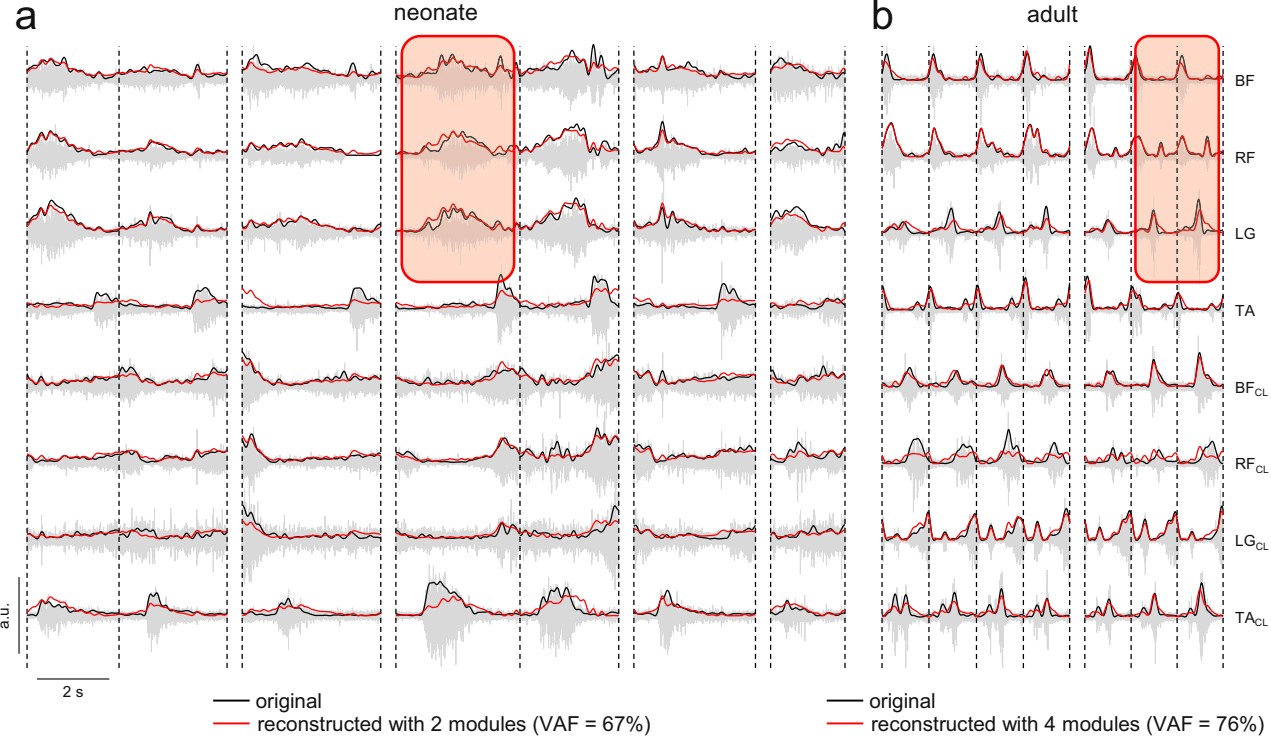

**Fig. 1 EMG activity of leg muscles during stepping. a** Raw EMG data from eight bilateral leg muscles (gray) in one representative neonate. Black lines represent the rectified EMG data, red lines represent the reconstructed EMG data using spatial decomposition with two modules. **b** Raw EMG data in one representative adult (same format as **a**). Red lines represent the reconstructed EMG data using spatial decomposition with four modules. Red areas highlight an example of the different inter-muscular coordination (synchronous in the neonate vs. asynchronous in the adult) of some groups of muscles. (BF Biceps Femoris, RF Rectus Femoris, LG Lateral Gastrocnemius, TA Tibialis Anterior, CL Contralateral Leg, VAF Variance Accounted For).

others peaks were uniformly distributed across frequencies from 0 to 10/T. Also, the peaks in the spectra of neonates had significantly lower power and larger bandwidth as compared to those of the adults (Fig. 2c, Wilcoxon rank sum test, $p < 0.001$ for both parameters).

Also the features of the aperiodic component[46], namely the corrected broadband offset, the corrected knee parameter and the exponent (see Methods), were significantly different between neonates and adults (Fig. 2d, Wilcoxon rank sum test, $p < 0.001$ for all parameters). In particular, the aperiodic component of the spectra of the EMG activity in neonates was significantly up-translated (by ~1 dB), had a narrower horizontal slope, and was less inclined past the knee inflection point compared to adults.

From the PSD we also calculated the spectral entropy (SE) to measure the degree of irregularity of the EMG signal of each muscle. SE was significantly higher in neonates for all analyzed muscles (Fig. 2e, Wilcoxon rank sum test, all $p < 0.001$), implying that EMG activity was less regular in the neonates.

Since irregularity over the seven concatenated strides may come in part from high variability in the stride duration in neonates, each stride duration was normalized to T for the analyses of Fig. 2. Nevertheless, even when we analyzed intra-muscular complexity of the original, unnormalized EMG data, we found higher complexity of EMG waveforms in adults compared to neonates (Supplementary Figure 1).

Another potential concern about the source of irregularity in neonates arises from analyzing strides that were not necessarily consecutive. However, as explained in Methods, to make all data sets as closely comparable as possible in all analyses, we randomly sampled seven strides of each adult (not necessarily consecutive) to generate the concatenated sequence of analyzed strides. In addition, we also performed the same frequency analyses on a

subset of seven neonates who produced five consecutive strides, and we obtained results similar to those in Fig. 2. In particular, the average number of peaks of the periodic component of the PSDs in neonates performing five consecutive strides was similar to that in neonates with seven, not necessarily consecutive strides; the average number of peaks over all muscles was $2.4 \pm 1.3$ (mean ± SD) and $3.3 \pm 1.1$, respectively (Wilcoxon rank sum test, $p = 0.18$ for BF, $p = 0.042$ for RF, $p = 0.053$ for LG, and $p = 0.28$ for TA). Moreover, this number for the neonates with five consecutive strides was significantly lower than in adults (Wilcoxon rank sum test, $p < 0.001$ for all muscles).

**Inter-muscular complexity in the spatial and temporal domain.** One way to assess the minimum complexity of inter-muscular coordination consists in using dimensionality-reduction algorithms, and assessing the dimensionality of the decomposition model that fits the EMG data of different muscles adequately. Typically, one increases the number of modules of the decomposition progressively, and computes the corresponding VAF. The minimum complexity is then defined as the minimum number of modules reaching a predefined VAF threshold. Alternatively, minimum complexity can be defined on the basis of the characteristics of changes in the VAF curve (such as its curvature or polynomial fitting). However, all these procedures and the resulting assessment of complexity may be affected by the presence of noise, especially if they are based on a single point on the curve (e.g. the threshold crossing point or the value for a single synergy). Therefore, here we combined traditional VAF measures with consistency measures that take into account the potential variability of motor modules across strides and participants.

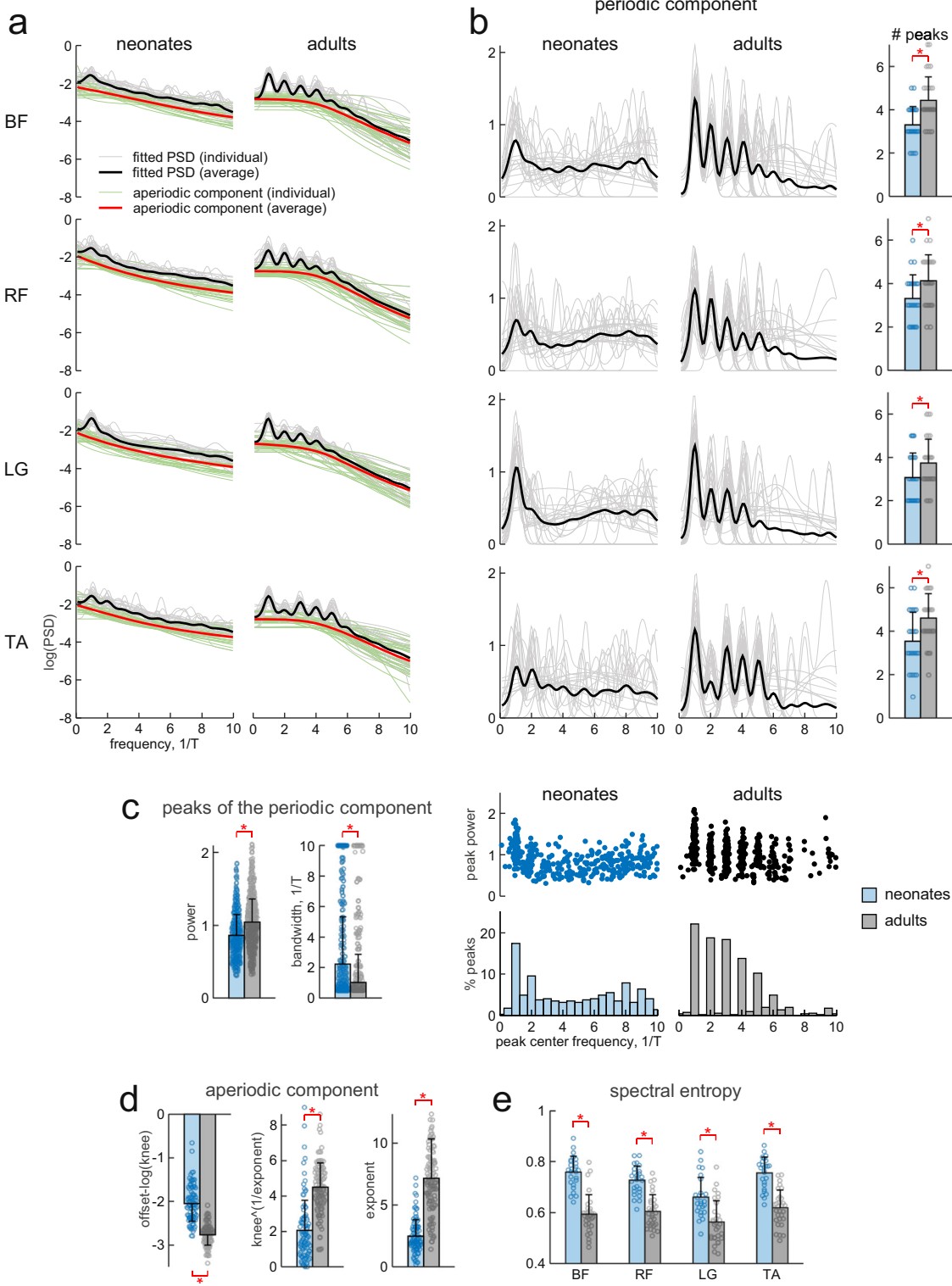

**Simulated data.** To assess the ability of these methods to capture the actual dimensionality of the EMG data in the presence of noise, we calculated the VAF by the reconstructed data and we also calculated three specific consistency measures (one for each decomposition method, see below). We tested these methods on different sets of simulated EMG data, which were obtained from predetermined modules (number of simulated modules ranging from 2 to 5) and contaminated by different levels of signal-dependent noise[48]. In addition, we used three different methods

of EMG factorization, spatial decomposition, temporal decomposition, and space-by-time decomposition[45] (Fig. 3a). We outlined the differences between these factorization methods in the Introduction. For further details, see Materials and Methods.

Figure 4 illustrates the results obtained using simulated EMG data sets originated from three modules and corrupted with increasing levels of noise (100 data sets for each noise level), but similar results were obtained from data sets involving different number of modules (Supplementary Figure 2). In Fig. 4, the r²

**Fig. 2 Parametrized frequency analysis of EMG data. a** Power spectral density (PSD) for all neonates (left) and all adults (right) was calculated from rectified EMGs interpolated over the gait cycle (T: stride duration) for each muscle (from top to bottom: BF, RF, LG, and TA, right and left leg muscles were pooled together) using Fast Fourier Transform (FFT) and fitted with an algorithm for parameterizing neural PSDs into periodic and aperiodic components[46]. Gray lines represent fitted PSDs from individual subjects, black lines represent average fitted PSD across subjects, green lines represent the aperiodic component from the individual PSDs, red lines represent the average aperiodic component across subjects. **b** Periodic component of the PSD for all neonates (left) and all adults (center), black lines represent average across subjects. The average (+SD) number of peaks (defined from the algorithm as the frequency regions of power over and above the aperiodic component) across subjects of each group are illustrated in the right column for each muscle ($n = 26$ for neonates and $n = 30$ for adults). **c** Characteristics of the peaks of the periodic component. The average (+SD) power (left) and bandwidth (right) of the peaks across subjects and muscles are shown on the left side of the panel ($n = 344$ for neonates and $n = 507$ for adults). The distributions of the peak power (upper plots, each point represents a peak, all peaks from all muscles of all subjects are pooled together) and of the percent number of peaks across center frequencies of the peaks are shown on the right side of the panel. Note that the great majority (~83%) of the peaks of the periodic component of the EMGs of adults was located near to the stride frequency ($1/T$) and its first four harmonics (from $2/T$ to $5/T$), while, in neonates, ~17% and ~10% of the peaks were located respectively around $1/T$ and $2/T$ and the others peaks were uniformly distributed across frequencies from 0 to $10/T$. **d** Parameters of the aperiodic component. From left to right average (+SD) corrected broadband offset, corrected knee, and exponent of the aperiodic component across subjects and muscles ($n = 104$ for neonates and $n = 120$ for adults). Note that all three parameters that characterize the arrhythmic activity of EMGs significantly differ between neonates and adults. **e** Average (+SD) spectral entropy across subjects for each muscle in neonates and adults ($n = 26$ for neonates and $n = 30$ for adults). Data points for all individuals are included in the histograms. Red asterisks denote significant differences between groups (Wilcoxon rank sum test $p < 0.05$).

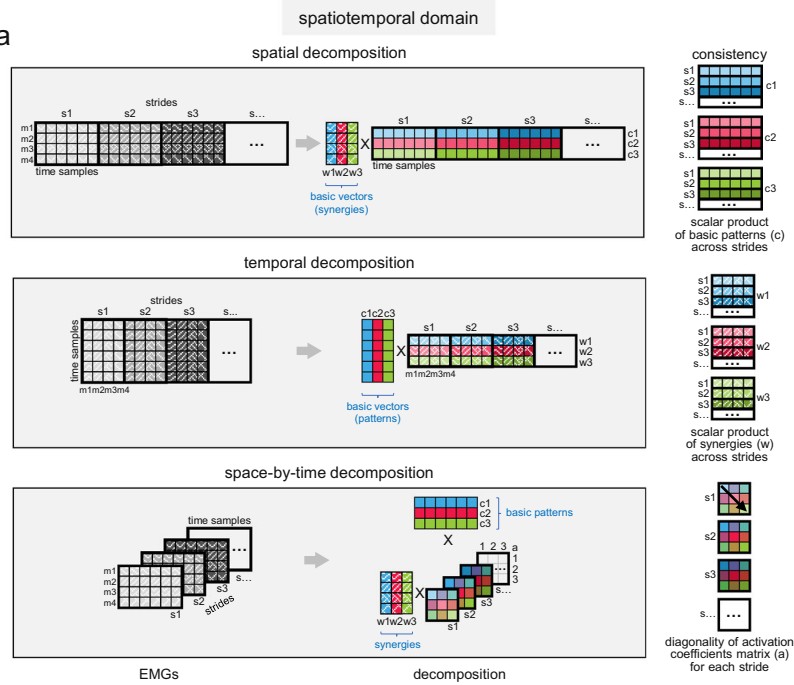

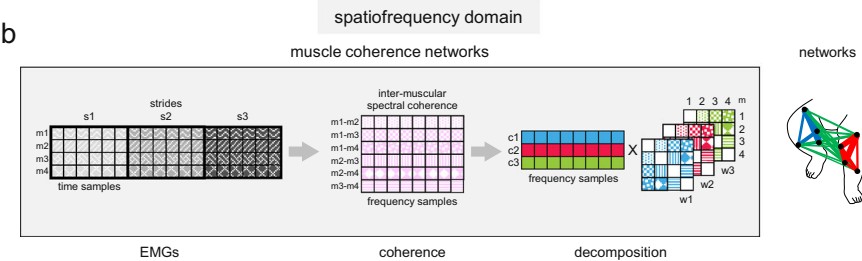

**Fig. 3 Factorization algorithms to assess the spatiotemporal and spatiofrequency organization of the muscle activity patterns. a** Type of multidimensional decomposition of EMG data (m) in the spatiotemporal domain: spatial decomposition, temporal decomposition, and space-by-time decomposition (from top to bottom). To assess consistent muscle modules across strides (s) the criteria of similarity was used: scalar product between basic patterns (c) for spatial decomposition, scalar product between synergies (w) for temporal decomposition, diagonality of the activation coefficients matrix (**a**) for space-by-time decomposition. **b** Muscle coherence networks for the multidimensional decomposition of EMG data in the spatiofrequency domain[47]. The inter-muscular coherence spectra of all muscle combination (m1-m2, m1-m3, …) are decomposed using non-negative matrix factorization. Each factor is characterized by the extracted feature (c) and the loadings of this feature in original spectra (w). These loadings give the strength of the edges between the nodes of each muscle network (right panel).

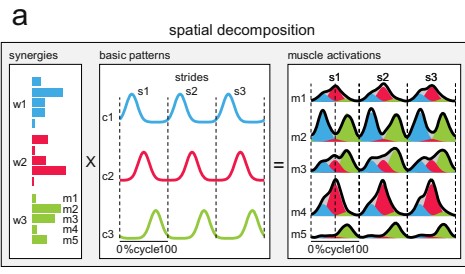

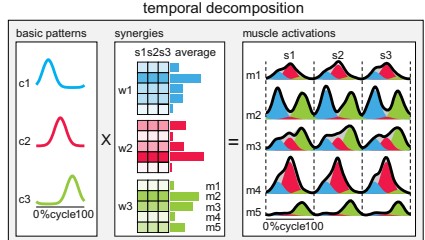

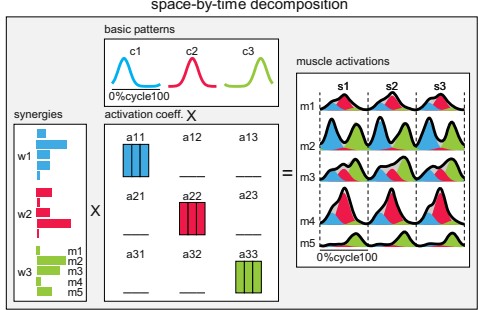

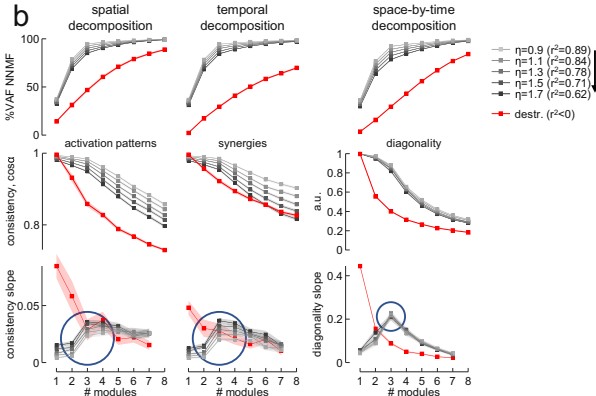

**Fig. 4 Effect of noise on simulated data. a** Schematic of motor modules using three different models (from top to bottom) spatial modularity ($m^s(t) = \sum_i c_i^s(t)w_i$), temporal modularity ($m^s(t) = \sum_i c_i(t)w_i^s$) and space-by-time modularity ($m^s(t) = \sum_i \sum_j c_i(t)a_{ij}^s w_j$). The outputs of the first (blue), second (magenta), and third (green) modules were summed together to generate an overall muscle activation ($m^s(t)$, black envelope). The simulated example of muscle activity profiles was computed using constant basic patterns ($c_i^s(t)$) for the spatial modularity model, constant synergies ($w_i^s$) for the temporal modularity model across different strides (s), and the identity matrix for the activation coefficients ($a_{ij}^s$) for the space-by-time modularity model. **b** Effect of noise on the percent of variance accounted for (VAF, upper) by the reconstruction of simulated EMGs using three different decomposition methods (from left to right, spatial, temporal, and space-by-time decomposition) and on the similarity measures (middle) and their slope (lower): average similarity of the activation patterns across strides for spatial decomposition, average similarity of synergies across strides for temporal decomposition and average diagonality of the activation coefficients matrix across strides for space-by-time decomposition (from left to right). The simulated EMGs were constructed, by an adaptation of the method proposed by Tresch et al.[48], using the corresponding equation for each model to calculate noiseless data value $m^s(t)$ and adding signal-dependent noise with $SD = \eta \cdot m^s(t)$, where $\eta$ is the slope of the relationship between the SD and noiseless data value. We constructed sets of 8 EMGs for 7 strides starting from 3 modules composed of constant synergies and basic patterns as in panel a. Each simulation was performed using increasing values of $\eta$ for 100 times (averages across simulations and confidence intervals—as shaded areas—are shown in the graphs). Note that VAF was significantly affected by noise, making it difficult to distinguish the exact number of modules ($N = 3$) used to generate the simulated EMGs for higher levels of noise for each decomposition method, while the similarity measures showed a substantial reduction going from 3 to 4 modules regardless of the level of noise. Red lines represent the results from structureless data obtained by randomly shuffling the simulated EMG samples (see Methods).

between noiseless and noisy EMG data ranged between 0.89 and 0.62 for $\eta$ ranging between 0.9 and 1.7. The VAF by assuming the same number of modules ($N = 3$) that generated the simulated data was significantly affected by the noise level for all decomposition methods. Thus, the VAF ranged from 94.7% ($CI_{95\%}$ [94.5%, 94.9%]) for $\eta = 0.9$ to 85% ($CI_{95\%}$ [84.6%, 85.6%]) for $\eta = 1.7$ for the spatial decomposition, from 94.7% ($CI_{95\%}$ [94.6%, 94.9%]) for $\eta = 0.9$ to 84.6% ($CI_{95\%}$ [84.2%, 85.1%]) for $\eta = 1.7$ for the temporal decomposition, and from 92.5% ($CI_{95\%}$ [92.1%, 92.8%]) for $\eta = 0.9$ to 78.7% ($CI_{95\%}$ [78%, 79.3%]) for $\eta = 1.7$ for the space-by-time decomposition (Fig. 4b). This means that using a fixed VAF threshold is not sufficient to estimate unambiguously the dimensionality of muscle activation modules in data sets affected by noise. Other criteria based on the VAF curves—such as their shape as a function of the number of modules—may also be inaccurate, since the curvature of VAF was also affected by the level of noise (Fig. 4b).

On the other hand, the shape of the three consistency measures was less affected by the amount of noise (Fig. 4b). We considered the consistency of basic activation patterns across strides for the spatial decomposition, the consistency of synergies across strides for the temporal decomposition, and the diagonality of the activation coefficients matrix for the space-by-time decomposition. Activation coefficients of the space-by-time decomposition represent the level of activation of each possible pair of spatial and temporal modules[45]. Consistency was computed as the average scalar product ($\cos\alpha$) of the basic patterns or the synergies across all possible pairs of strides. This scalar product estimates the similarity of a given pattern or synergy across strides. Diagonality was computed as the average ratio of the diagonal activation coefficients to all activation coefficients (Eq. 6 in Methods). We then considered how these consistency measures change as a function of the number of modules (from 1 to 8), by taking the slope of this function (lower panels in Fig. 4b). Maximum slope locates the point of the function where increasing or decreasing the number of modules relative to this point leads to the most drastic change of the corresponding consistency parameter. Thus, the slope represents a sensitive indicator for the dimensionality of modules in the presence of noise. In particular, the slope of the diagonality of the space-by-time decomposition showed a clear peak corresponding to the actual number of simulated modules ($N = 3$), which remained stable over a wide range of noise levels (Fig. 4b, lower panels). Similar results were obtained with different numbers of simulated modules (Supplementary Figure 2). Notice that structureless data obtained by randomly shuffling the simulated EMG samples

(see Methods) did not exhibit the same trend as the original data (red points in Fig. 4b).

In the simulations of Fig. 4 and Supplementary Figure 2, the underlying basic patterns (for spatial decomposition), synergies (for temporal decomposition), or activation coefficients (for space-by-time decomposition) that generated the EMGs were constant across the strides before being corrupted by different levels of noise. In a different set of simulations, we relaxed this constraint and assessed the effect of cycle-to-cycle variability of either the timing (Supplementary Figure 3a) or the amplitude (Supplementary Figure 3b) of the basic patterns on the consistency measures. In Supplementary Figure 3, the $r^2$ between noiseless and noisy EMG data ranged between ~0.7 and 0.4 for $\eta$ ranging between 0.9 and 1.7. We found that, also in these simulations, the slope of the consistency measures exhibited a peak (especially marked for the diagonality of the space-by-time decomposition) corresponding to the actual number of simulated modules ($N = 3$), which remained stable over a wide range of noise levels.

*Experimental data.* Biological data, and in particular EMG signals, are affected by different types and levels of noise[49]. In fact, the VAF curves calculated from the actual EMG data of neonates, infants, and adults as a function of different number of modules for different decomposition methods (principal component analysis PCA, and non-negative matrix factorization NNMF) showed the same features as those obtained from the simulated data affected by high levels of noise (Fig. 5a). For a number of modules less than eight (the maximum number that is theoretically possible, given that there are eight muscles), VAF was higher in neonates and decreased with age in infants (a trend especially clear with the spatial decomposition model, Fig. 5a), consistent with an increasing dimensionality with age. However, the differences across age groups were small and insufficient to assess the dimensionality of the data sets unambiguously.

Therefore, for each participant we also evaluated the number of modules corresponding to the maximum slope of the specific measures of consistency across strides that we used with the simulated data (Fig. 5b). The number of modules identified by these criteria was variable across participants of all age groups, including the adults (Fig. 5c). In particular, one can notice in Fig. 5c that a few participants of all age groups, except the adults, showed only one module of neuromuscular activity. This was due to low-frequency oscillations of ipsilateral EMG activities that predominated on all other components. Overall, the consistency measures showed a systematic trend toward increasing dimensionality of the modules with increasing age. Figure 5d plots the mean values of the modules identified using the maximum of the slope of diagonality of the activation coefficients matrix (space-by-time decomposition) versus the mean age of each group of participants. The trend with age was statistically significant: the linear regression of the number of modules versus age yielded $r = 0.88$ including only all children, and $r = 0.95$ including also the adults. Very similar results were obtained using the number of modules identified from the consistency measures for the spatial and temporal decomposition.

Notice that the VAF by two modules in neonates was significantly lower than the VAF by four modules in adults for all decomposition models, presumably due to higher noise (unstructured variability) in the former than in the latter participants. In neonates, two modules explained on average 62% ($CI_{95\%}$ [59%, 66%]) of the variance of the EMG data with the spatial decomposition, 44% ($CI_{95\%}$ [41%, 47%]) with the temporal decomposition, and 41% ($CI_{95\%}$ [39%, 44%]) with the space-by-time decomposition. In adults, 4 modules explained on average 83% ($CI_{95\%}$ [81%, 84%]) of the variance of the EMG data for

spatial decomposition, 81% ($CI_{95\%}$ [77%, 83%]) for temporal decomposition, and 78% ($CI_{95\%}$ [75%, 81%]) for space-by-time decomposition. Critically, however, the number of modules necessary to account for a given level of variance tended to increase with age, consistent with the hypothesis that the dimensionality of neuromuscular control increases with age.

In addition to the quantitative parameters, it is helpful to assess qualitatively the ability of the decomposition methods to describe the original data. Figure 1 reports examples of data reconstruction in one neonate and one adult using the spatial decomposition algorithm with two and four modules, respectively. In general, the envelope profiles of EMG activities are reproduced reasonably well by the algorithm, but there were also episodes of poor correspondence. The results from all neonates (except participant N6, see Methods) and adults using the spatial decomposition and the temporal decomposition are plotted in Supplementary Figure 4 and Supplementary Figure 5, respectively, with two and four modules in neonates and 4 modules in adults.

The same data are analyzed using the space-by-time decomposition in Fig. 6. This method, in particular, allows a clear visual assessment of how changing the number of dimensions would affect the way one can account for the data in the presence of variability[45]. In particular, the matrix of the activation coefficients for all single strides indicates the extent to which the participants of a given age group use the same modules (patterns and synergies) in all strides. Thus, higher values of the activation coefficients along the matrix diagonal ($a_{ii}$) relative to the activation coefficients off-diagonal ($a_{ij}$ and $a_{ji}$) denote a greater consistency of engagement of the same modules in most strides across participants of a given age group. Figure 6 shows that, in neonates, this is the case for two modules but not for four modules, consistent with our previous quantitative assessment. Indeed, it can be noticed that the coefficients $a_{11}$ and $a_{22}$ are much higher than $a_{12}$ and $a_{21}$ for two modules in neonates (Fig. 6a), whereas $a_{11}$, $a_{22}$, $a_{33}$, and $a_{44}$ are only slightly higher than the other coefficients for 4 modules (Fig. 6b). By contrast, $a_{11}$, $a_{22}$, $a_{33}$, and $a_{44}$ are much higher than the other coefficients for 4 modules in the adults (Fig. 6c).

As an additional approach to investigate the dimensionality of the neuromuscular modules, we used the cluster analysis to identify similar muscle synergies and activation patterns across all recorded strides of all participants of each age group[36]. To this end, we first applied NNMF to each single stride of each participant separately, and we retained the smallest number of modules that accounted for ≥80% of the variance of EMG profiles. Next, we clustered individual muscle synergies and activation patterns, and found that the optimal number of clusters for both sets of variables was two and four in neonates and adults, respectively (Supplementary Figure 6). In neonates, 82% ($n = 277$) of all activation patterns and 76% of all muscle synergies were above the silhouette threshold (see Methods). In adults, 90% ($n = 399$) of all activation patterns and 85% of all muscle synergies were above the silhouette threshold. The timing and shape of these patterns, as well as the values of the associated muscle synergies obtained with cluster analysis were very similar to those plotted in Fig. 6a for neonates and Fig. 6c for adults.

In sum, different approaches converge toward the conclusion that neonates employ fewer modules of neuromuscular control during stepping than adults. However, since we could not account for a considerable fraction of the data variance in neonates, one may wonder whether there was any systematic structure in the residuals that were not fit by the decomposition models. To address this issue, we computed the similarity (scalar product) between the residuals in all muscles and neonates or adults from the space-by-time decomposition (similar results were obtained using the other decomposition methods). We found that the

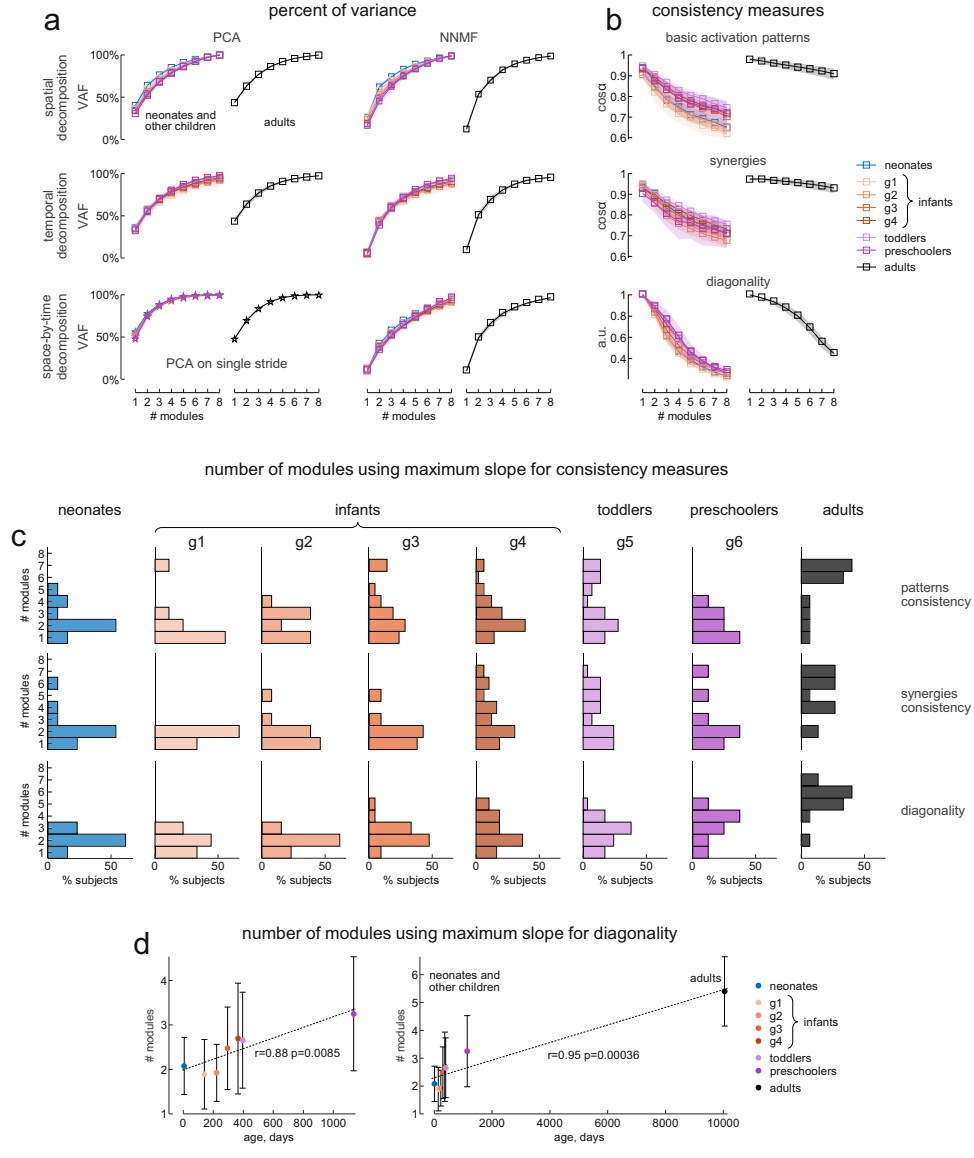

**Fig. 5 Percent of variance accounted for and inter-stride consistency of experimental reconstructed data in neonates and the other groups. a** Average (z-transformed, across subjects) percent of variance accounted for (VAF) by the reconstructed EMGs in infants (left panels) and adults (right panels) using different number of modules and different decomposition algorithms and methods. In the left columns using principal component analysis (PCA) algorithm, from top to bottom: spatial decomposition, temporal decomposition, and PCA on single stride. In the right columns using non-negative matrix factorization (NNMF) algorithm, from top to bottom: spatial decomposition, temporal decomposition and space-by-time decomposition. Shaded areas denote 95% confidence intervals across subjects. Note that VAF seemed to be higher in neonates and to decrease with age in infants, though it was not sufficient to evaluate the dimensionality of the data sets. **b** Average (z-transformed, across subjects) inter-stride consistency measures across subjects for the modular decomposition (NNMF) of bilateral EMGs using different number of modules in infants (left column) and adults (right column). From top to bottom: inter-stride consistency of basic activation patterns (calculated as cosα, where α is the angle between two basic activation pattern vectors) derived from spatial decomposition, inter-stride consistency of synergies (cosα) derived from temporal decomposition, diagonality of the activation coefficient matrices derived from space-by-time decomposition. Shaded areas denote 95% confidence intervals across subjects. **c** Histograms, for each group of subjects, of the number of modules identified using the maximum of slope of the different consistency methods (from top to bottom as in **a**). Bar height denotes the percentage of subjects whose maximum slope of consistency measures is located in the corresponding number of modules. **d** Correlation between the number of modules identified from the maximum slope of diagonality and the age in days for each group of participants, excluding (left panel) and including (right panel) adults. Each data point represents the average (±SD) value across subjects for each group. Dashed lines, linear regressions. The sample size is the same for all panels ($n = 13$ for neonates, $n = 9$ for infants g1, $n = 13$ for infants g2, $n = 21$ for infants g3, $n = 49$ for infants g4, $n = 29$ for toddlers, $n = 8$ for preschoolers, and $n = 15$ for adults).

similarity did not differ significantly from 0 (corresponding to complete dissimilarity) for any number of modules greater than two in neonates ($CI_{95\%}$ encompassing 0), whereas in adults the similarity differed significantly from 0 for any number of modules. Two-way ANOVA showed that the similarity of residuals was significantly higher in adults than neonates [main

effect of group, $F_{(1,1776)} = 1611.00$ $p < 0.001$], decreased significantly with increasing number of modules [main effect of number of modules, $F_{(7,1776)} = 217.71$ $p < 0.001$], and was significantly lower in neonates than adults with increasing number of modules [interaction effect, $F_{(7,1776)} = 101.37$ $p < 0.001$]. Therefore, in neonates all EMG activity not accounted

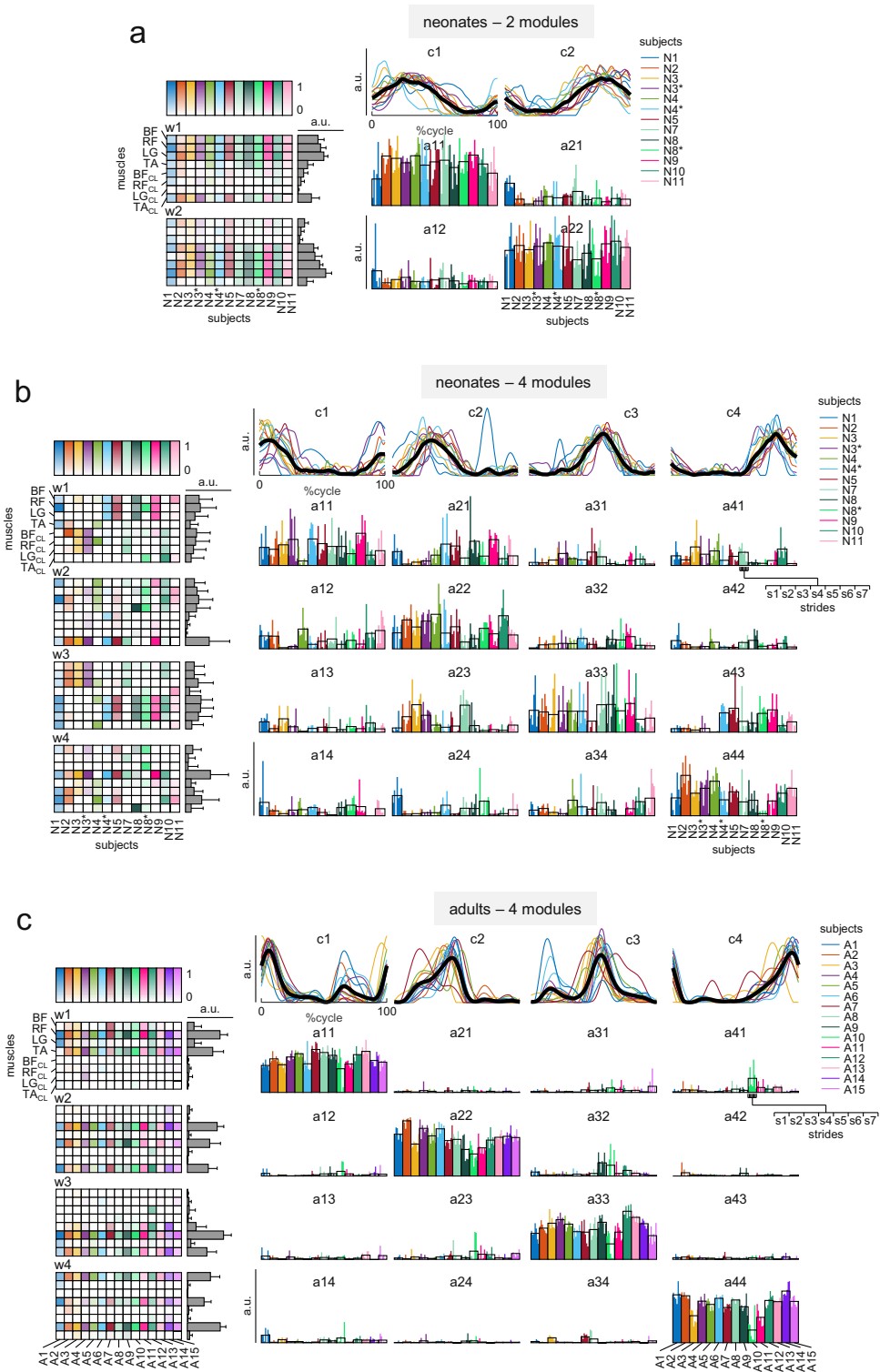

for by two spatiotemporal modules reflects unstructured step-by-step variability (noise). By contrast, in adults one may still find significant structure even in higher order components.

*Functional comparison of the modularity between neonates and adults.* Figure 6a, Supplementary Figure 5a and 6b show that the two main patterns of EMG activity in neonates exhibit sinusoidal-like waveforms, the first pattern (c1) peaking at ~30% of the cycle (midstance), and the second one (c2) peaking at ~75%

(midswing). On each limb, c1 recruits the quasi-synchronous activation of several extensor and flexor muscles, contributing to stiffen the limb and exerting vertical forces supporting part of body weight[32,34,36,42]. Neonates typically support ~30–40% of their weight during this stepping phase[43,50]. c2 recruits mainly flexor muscles such as tibialis anterior, contributing to flex the leg and foot. Neonates typically lack major muscle activity at either touch down or lift-off, and they show correspondingly small tangential forces at the step-by-step transitions[32]. Forward progression/propulsion is

**Fig. 6 Space-by-time decomposition. a** Result of the space-by-time decomposition of EMG data of neonates using two modules. Left column: muscle synergies ($w_1$ and $w_2$, from top to bottom), each column represents a single neonate, and each row represents a muscle (BF: Biceps Femoris, RF: Rectus Femoris, LG: Lateral Gastrocnemius, TA: Tibialis Anterior, CL: Contralateral Leg), the intensity of the color is proportional to the muscle weight, the average (+SD) values are illustrated through gray bars on the right. Upper row: basic activation patterns ($c_1$ and $c_2$, from left to right), each colored line represents a single neonate, black lines indicate the average across subjects. Middle panels: elements of the activation coefficients matrix ($a_{11}$, $a_{12}$, ..., $a_{22}$), each bar represents a single stride ($s_1$-$s_7$) of each neonate (in different colors). The modules of each subject are grouped in order to attain the minimum distance (1-scalar product) between the basic activation patterns of all subjects. Since for the space-by-time decomposition there is not a unique relationship between synergies and basic patterns, which are related by the NxN matrices of activations coefficients, after sorting the basic activation patterns, we sorted also the muscle synergies in order to obtain the resulting activation coefficient matrix as close as possible to a diagonal matrix. **b** Space-by-time decomposition of EMG data of neonates using four modules, same format as panel a. **c** Space-by-time decomposition of EMG data of adults using four modules, same format as panels a and b. Note that, despite four modules in neonates account on average for 69% ($CI_{95\%}$ [66%, 71%]) of the variance of the EMG data [in neonates two modules account on average for 41% ($CI_{95\%}$ [39%, 44%]) of the variance, and in adults four modules account on average for 78% ($CI_{95\%}$ [75%, 81%]) of the variance], these modules are visibly less consistent across strides and across subjects (the average synergies across subjects are flatter and the non-diagonal elements of the activation coefficients matrices are noticeably higher) compared with the decomposition with two modules in neonates (a) and four modules in adults (panel c). N6 data are missing because this neonate performed 5 consecutive strides, but did not have the seven strides necessary for factor analysis (see Methods).

generally provided by the experimenter (or treadmill belt) rather than by the neonate. Overall, the sequence of muscle activations generates the idiosyncratic style of locomotion of neonates, involving ground contact with variable parts of the foot sole, and hyperflexion of the lower limbs during swing[32,34,50].

Figure 6c, Supplementary Figure 5c and Supplementary Figure 6b show that adults exhibit 4 main patterns of EMG activity, each of much shorter duration relative to the neonatal patterns. On each limb, these patterns are accurately timed around the four critical events of the gait cycle, heel strike, weight acceptance/forward propulsion, lift-off, and touch down. Adults show a much lower extent of muscle co-contraction as compared with neonates and infants[42,43]. During stance, the limbs are kept relatively extended, and the center of pressure on the ground shifts smoothly heel-to-toe[34].

*Inter-subject consistency.* The inter-subject consistency of the EMG modules extracted using different decomposition methods decreased significantly with increasing number of modules (Fig. 7a, two-way ANOVA $F_{(7,1192)} = 1452.82$ $p < 0.001$, $F_{(7,1192)} = 500.64$ $p < 0.001$, and $F_{(7,1192)} = 416.66$ $p < 0.001$ for the effect of the number of modules for basic activation patterns extracted respectively with spatial, temporal and space-by-time decomposition and $F_{(7,1192)} = 856.21$ $p < 0.001$, $F_{(7,1192)} = 1378.52$ $p < 0.001$, and $F_{(7,1192)} = 694.90$ $p < 0.001$ for the effect of the number of modules for muscle synergies extracted respectively with spatial, temporal and space-by-time decomposition). Also, the decrease of inter-subject consistency with increasing number modules was significantly different across age groups (Fig. 7a, two-way ANOVA $F_{(49,1192)} = 4.77$ $p < 0.001$, $F_{(49,1192)} = 6.10$ $p < 0.001$, and $F_{(49,1192)} = 6.02$ $p < 0.001$ for the interaction between the effect of the number of modules and the effect of group for basic activation patterns extracted respectively with spatial, temporal and space-by-time decomposition and $F_{(49,1192)} = 12.86$ $p < 0.001$, $F_{(49,1192)} = 8.37$ $p < 0.001$, and $F_{(49,1192)} = 13.26$ $p < 0.001$ for the interaction between the effect of the number of modules and the effect of group for muscle synergies extracted respectively with spatial, temporal and space-by-time decomposition). In particular, considering a fixed number of modules ($N = 4$) for all groups of participants, the inter-subject variability of the basic activation patterns obtained using spatial decomposition and of the muscle synergies extracted with all decomposition methods was higher (meaning less consistent modules across participants) in infants compared with adults (Fig. 7b, Tukey's Honestly Significant Difference test, $p < 0.047$ for all the comparisons shown in the Figure).

**Inter-muscular complexity in the frequency domain.** The coordination between multiple muscles can be assessed also in the frequency domain by studying the characteristics of the muscle networks derived from the spectral coherence between different pairs of muscles[47]. We applied this approach to the EMG data of neonates and adults in order to evaluate the complexity of multi-muscle control in the frequency domain for these two groups of participants.

Consistent with the results obtained in the spatiotemporal domain, the dimensionality of the muscle networks (assessed by the VAF curves calculated from the decomposition of the inter-muscular coherence spectra with different number of modules) was smaller in neonates than in adults (Fig. 8a). In fact, on average $2.5 \pm 1.7$ ($\pm$SD) modules with PCA and $3.2 \pm 1.9$ modules with NNMF were sufficient to reach a VAF $\geq 80\%$ of the reconstructed coherence spectra in neonates, while $3.3 \pm 1.7$ modules with PCA and $4.4 \pm 2$ modules with NNMF were required in adults to reach the same VAF threshold (Fig. 8b).

In Fig. 9, we illustrate the features of the inter-muscular coherence networks obtained with two components in neonates (accounting on average for 79% of variance, $CI_{95\%}$ [69%, 86%], Fig. 9a), and with four components in adults (accounting on average for 82% of variance, $CI_{95\%}$ [78%, 86%], Fig. 9b). The networks of different participants from each group were clustered according to the between-subjects similarity (scalar product) of the frequency components ($c_1$-$c_2$ in Fig. 9a and $c_1$-$c_4$ in Fig. 9b), and sorted by the frequency of the main peak of these components. Only the frequency components of the first network in neonates ($c_1$ in Fig. 9a), and the frequency components of the first three networks in adults ($c_1$-$c_3$ in Fig. 9a) were consistent across participants of the same group. This means that common fluctuations of muscle EMGs in neonates have coherence components mainly at low frequencies (below 5 Hz), while in adults, in addition to the low-frequency components, there are distinct components ranging up to ~12 Hz (Fig. 9). Supplementary Figure 7 illustrates similar results obtained using one component of the coherence spectra in neonates and three components in adults. Moreover, the topological features (anatomical distribution across different sets of muscles) of all extracted networks were significantly different (Kruskal–Wallis $p < 0.001$ between neonates and adults for betweenness-centrality and average edge weight, Fig. 9c). Even the first networks of neonates and adults ($n_1$ in Fig. 9) were quite different in topology, despite their similar spectral signature. The muscular network $n_1$ in neonates was characterized by higher (even if not statistically significant, $p = 0.07$) betweenness-centrality and significantly lower (Tukey's Honestly Significant Difference test, $p = 0.0017$) interlimb average weight as compared to $n_1$ in adults, implying a weaker bilateral connectivity of the muscle network of neonates.

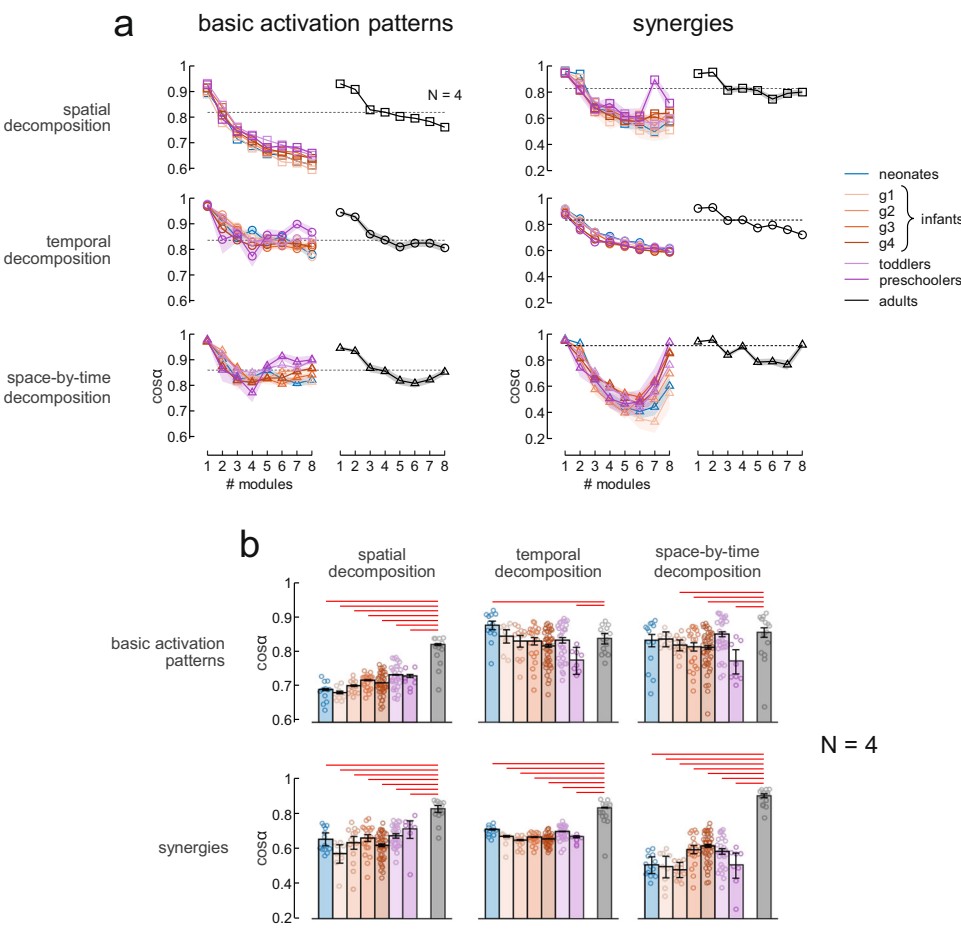

**Fig. 7 Inter-subject consistency of EMG modules in neonates and the other groups. a** Average inter-subject consistency measures across all possible couples of subjects in each group for basic activation patterns (left) and synergies (right) using different number of modules in infants (left panels) and adults (right panels) derived using different decomposition methods. From top to bottom: spatial decomposition, temporal decomposition and space-by-time decomposition. Shaded areas denote 95% confidence intervals across all possible couple of subjects. Dashed lines indicate the average consistency between subjects for adults using decomposition models with $N = 4$ modules. **b** Detail of the average (±95% confidence interval) inter-subject consistency measures across all possible couples of subjects for basic activation patterns (top) and synergies (bottom) using decomposition models with $N = 4$ modules. From left to right: spatial decomposition, temporal decomposition, and space-by-time decomposition. Note that inter-subject variability is higher (less consistent modules) in infants if compared with the same number of modules as in adults ($n = 13$ for neonates, $n = 9$ for infants g1, $n = 13$ for infants g2, $n = 21$ for infants g3, $n = 49$ for infants g4, $n = 29$ for toddlers, $n = 8$ for preschoolers and $n = 15$ for adults). Data points for all individuals are included in the histograms. Horizontal red lines denote significant differences between groups (post-hoc Tukey–Kramer multiple comparison $p < 0.05$, only significant differences between neonates and infants compared to adults are reported).

In agreement with previous reports[47], we found that muscle connectivity in adults takes place via long-range connections (e.g., the bilateral connections between legs), especially in the low-frequency muscle network. The new finding is that, in neonates, the low-frequency muscle network is characterized by shorter-range connections (mainly restricted to intra-limb connections), as shown by the lower average weight of the interlimb edges (Fig. 9c, lower right panel) and by the higher betweenness-centrality (Fig. 9c, upper right panel). Betweenness-centrality quantifies the importance of individual nodes of the network to keep interconnections. The nodes of neonatal networks have higher centrality due to the lack of long-range (interlimb) pathways, implying weaker or less coordinated bilateral inter-muscular networks.

## Discussion
It has previously been argued that the high intra- and inter-individual variability of muscle activities typical of infant stepping

indicates the lack of any underlying structured pattern, thus refuting the hypothesis of innate modular control[26,27]. Indeed, it may be difficult to determine putative neuromuscular modules and their dimensionality in the presence of noise[44,48]. Here, we tackled the potential confounds due to noise and structured variability by means of different approaches. First, we simulated several data sets with varying amount of noise, matching real data but with known dimensionality. We used 3 different algorithms for data factorization, namely spatial decomposition, temporal decomposition, and space-by-time decomposition, each making different assumptions[45]. We observed that, in the presence of noise, a criterion based on a given VAF threshold may not suffice to recover the dimensionality of the simulated data. However, consistency measures, and in particular the slope of the diagonality of the space-by-time decomposition algorithm, identify the dimensionality of noisy data unambiguously (Fig. 4b). Next, we applied the same decomposition algorithms to the experimental data of a cohort of children and

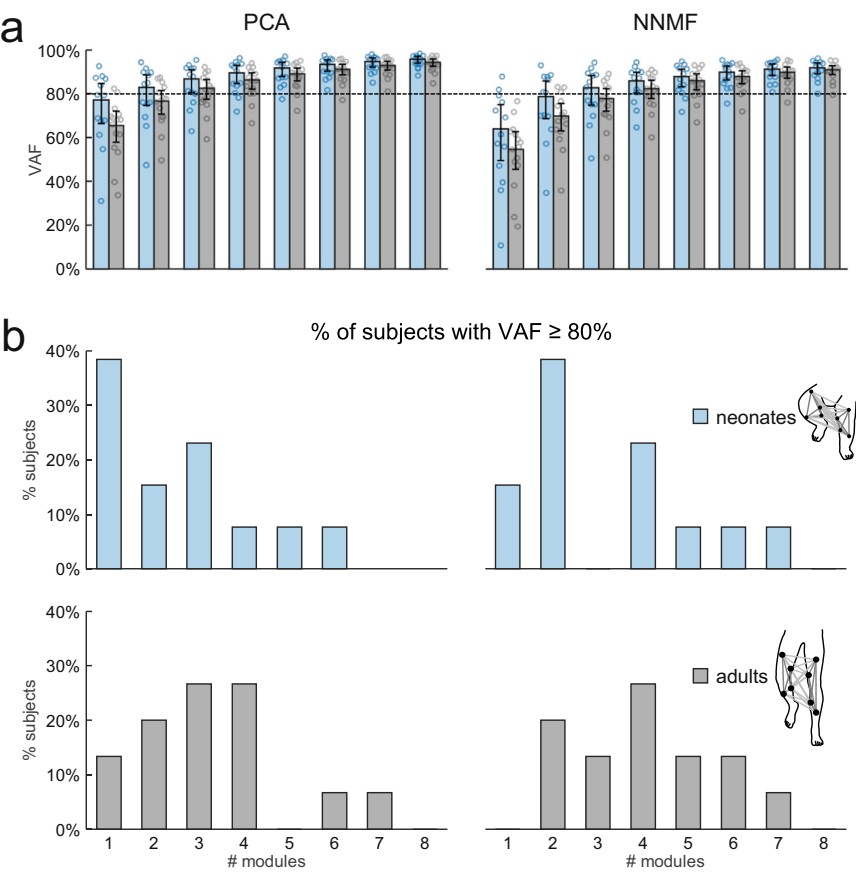

**Fig. 8 Dimensionality of muscle coherence networks. a** Average (z-transformed, across subjects) percent of variance accounted for (VAF) the reconstruction of the inter-muscular coherence spectra between all muscle pairs in neonates ($n = 13$) and adults ($n = 15$) using different number of modules and different decomposition algorithms: principal component analysis (PCA, left) and non-negative matrix factorization (NNMF, right). Vertical error bars denote 95% confidence intervals across subjects. Data points for all individuals are included in the histograms. **b** Histograms of the number of modules that account for at least 80% of the VAF in neonates (upper row) and adults (lower row) using the two different algorithms (PCA left and NNMF right).

adults, and found that the number of modules estimated by our consistency measures increased with age (Fig. 5c, d). The number of modules was variable across participants of all age groups, including adults. Also previous studies found inter-subject variability in the number of locomotor modules extracted by factorization methods in adults[16,18,19] and children[51]. However, both the basic activation patterns and the muscle synergies exhibited much greater inter-subject variability in all children groups compared with the adults (Fig. 7).

Overall, our decomposition methods concurred in showing that (i) neuromuscular modules are identifiable in all age groups, including the newborns, (ii) the number of modules increases with age, (iii) intra- and inter-individual variability exists in all age groups, but it is considerably higher in neonates and infants than in adults. Therefore, modularity coexists with variability, but while the complexity of modular organization increases with age, the associated variability decreases.

We also showed that the increase in complexity and decrease of variability from the neonatal stage to adulthood is evident in the frequency content and entropy of the individual EMGs (Fig. 2). Thus, the frequency peaks in neonates were significantly less numerous, had lower power and larger bandwidth as compared to adults. Also, the aperiodic component of the frequency content and the spectral entropy were significantly different between

neonates and adults, consistent with a greater irregularity of the intrinsic rhythmicity of EMG activity in neonates.

Complementary findings emerged by considering the spectral coherence between pairs of muscles[47]. Thus, the muscle networks of neonates had significantly lower dimensionality (number of modules of the inter-muscular coherence spectra) and weaker bilateral connections than in adults (Figs. 8–9). Moreover, inter-muscular coherence in neonates showed only low-frequency (<5 Hz) components, which are deemed to be associated with common modulation of motor unit firing rate and muscle force generation[52]. By contrast, in adults we found additional coherence components at higher frequencies (up to ~12 Hz), which may reflect supraspinal inputs on spinal motoneurons[53].

In this study, we recorded the EMG activity of eight muscles, rectus femoris, biceps femoris, tibialis anterior, and gastrocnemius lateralis of both lower limbs. These muscles were selected because they correspond to those analyzed in several previous studies in neonates and infants[27,32,33,37,42]. While a larger set of muscles would allow a more detailed description of neuromuscular control of locomotion, in a previous study[36] we showed that the basic activation patterns did not differ appreciably when extracted from the present set of muscles ($n = 8$) or a larger set of muscles ($n = 22$). Therefore, we believe that the main conclusions of our work would not change by including a larger set of muscles.

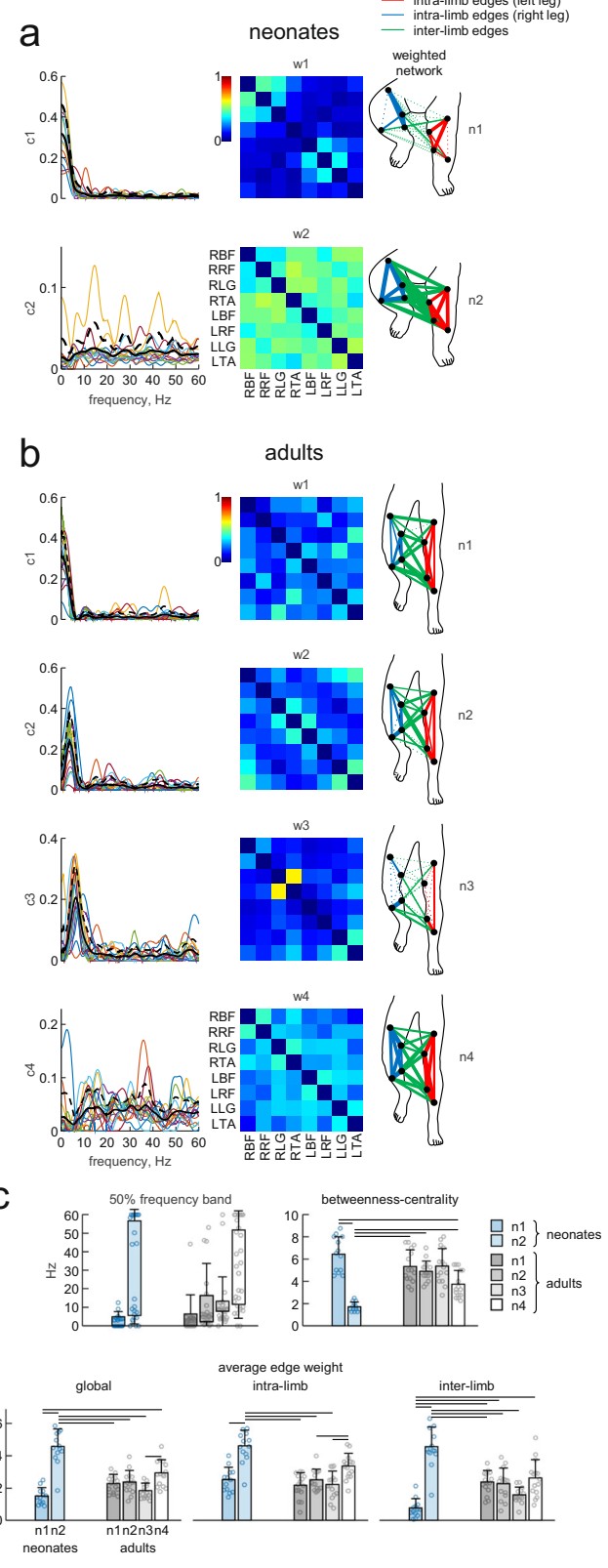

**Fig. 9 Inter-muscular coherence networks for neonates and adults. a** First two components (accounting on average 79% of VAF, 95% confidence interval [69%, 86%]) obtained with non-negative matrix factorization of the frequency content of the coherence spectra of all muscle combinations in neonates ($n = 13$). The components are ordered (from top to bottom) based on the peak of the frequency components (c1–c2, left column, black lines denote average across subjects +SD in dashed black lines). The central column shows the average loadings (w1–w2) of the corresponding frequency component across subjects. These loadings give the strength of the edges between the eight nodes of each muscle-weighted network (n1–n2, right column), connection strength is reflected by the width of the lines. **b** First four components (accounting on average 82% of VAF, 95% confidence interval [78%, 86%]) obtained with non-negative matrix factorization of the frequency content of the coherence spectra of all muscle combinations in adults ($n = 15$, same format as a). **c** Summary statistics of inter-muscular coherence networks. Network metrics were used to statistically compare the muscle networks across groups and frequencies: average (±SD) 50% frequency band across subjects (upper row, left panel), average (+SD) betweenness-centrality across subjects (upper row, right panel), and average (+SD) edge weight across subjects (lower row) considering all network edges (global, left panel), only the intra-limb edges (intra-limb, central panel) or only the interlimb edges (interlimb, right panel). Data points for all individuals are included in the histograms. Horizontal black lines denote significant differences between networks (post-hoc Tukey-Kramer multiple comparison $p < 0.05$).

behavior are laid down in the spinal cord at early stages of development, well before birth[54,55]. However, they are then modified extensively over a prolonged postnatal period by experience, with spontaneous neural activity, sensory feedback, and supraspinal control playing a critical role in shaping progressively the locomotor function[24,40,41,56]. Similarly, developmental studies in humans show that locomotor-like behavior is present prior to and at birth[25,32–36,42], but the kinematic and kinetic features typical of mature locomotion are reached only after several years[57].

There is increasing evidence that both rodents and primates (as well as other altricial species) are born with immature nervous systems involving circuits that have neither the neuronal properties nor the connectivity needed for future behaviors[40,58]. Neuronal properties and connectivity are tuned to limb and body biomechanics during the critical period of postnatal development[59]. At birth, several properties of the neuromuscular system differ profoundly from the mature stage, and in line of principle might account for the present results in neonates. Thus, although muscle fiber specification (type I or II, based on ATPase histochemistry) takes place in utero, physiological studies in neonatal cats show that both fast and slow muscles initially exhibit similarly slow contraction, which speeds up weeks after birth[60]. Also the conduction velocities of motor and sensory peripheral nerves in neonates are about 3 times slower than in adults. Prior to birth, motor axons establish widespread multiple innervation of muscle fibers that remains until several weeks after birth in both rodents and humans[61]. This polyinnervation might contribute to the muscle coactivation and low-dimensional control we found in neonates. The biophysical properties of mouse spinal motoneurons also change drastically after birth, maximal firing increasing and excitability decreasing into the third postnatal week[58]. A high-density neuromuscular interface in kicking human neonates showed relatively low discharge rates of spinal motoneurons, which were not modulated with increasing muscle contraction speed but were strongly synchronized across motoneurons[52]. Motoneuron synchronization had previously been observed in neonatal rodents, and associated to

In the following, we discuss the possible underpinnings of our findings.

**Neural and biomechanical determinants of developmental changes in neuromuscular control.** Work in animal models shows that the basic neural circuits underlying locomotor-like

motoneuron path-finding, synapse maturation (e.g., establishing the initial polyneuronal innervation and subsequent synapse elimination), and refinement of pattern-generating circuits[62]. Motoneuron synchronization is indicative of strong common synaptic input from premotor interneurons of locomotor pattern generators and from afferent feedback[53], and supports the modular neuromuscular organization we reported here also in neonates. In addition, transient electrical gap-junction coupling between motor neurons may account for motoneuron synchronization in neonates[62]. All these factors may contribute to the extensive muscle coactivation and the resulting low-dimensionality of neuromuscular modules in neonatal stepping.

Rhythmic bursting patterns of motoneurons are revealed in embryonic and fetal rodent preparations, progressively evolving from mainly synchronous to mainly alternating patterns between contralateral limbs[55]. Intrinsic rhythmogenesis can be revealed also in the isolated spinal cord of neonatal rodents stimulated by different neurotransmitters[29], but these rhythms are often quite irregular and variable[63]. The irregular rhythmicity revealed by the present frequency analysis of EMG activities in individual muscles of neonates appears consistent with these observations in neonatal animals.

Left-right coordination matures with different time courses in different animal species[55]. Here, it was assessed quantitatively by means of muscle networks analysis[47]. We found that the networks of neonates had lower dimensionality and weaker bilateral connectivity than adults' networks. This indicates that locomotor pattern generators can operate independently for the left and right lower limbs in neonates, and that the interlimb coupling is weaker than in adults[64]. A weak interlimb coupling partly depends on the fact that cortical control contributes little to neonatal stepping[65]. Interlimb coupling is weaker in the absence of cortical descending inputs, as shown for instance in spinal cats[66].

Weak interlimb coupling of neonates may also depend on immature sensory feedback from load signals and hip-position signals[24,67,68]. Sensory feedback may also contribute to shape neuromuscular modularity, expanding or compressing the number of modules. We previously compared the results of factorization of neonatal EMG activities during ground stepping and spontaneous kicking[36]. While neonatal stepping is triggered by the contact with the support surface and involves strong sensory signals about limb load and hip extension[24,67,68], sensory inputs are not necessary for triggering spontaneous kicking movements, which involve limited feedback about limb load and hip extension. We found that kicking involves activation patterns with a similar dimensionality and waveform as those of more mature locomotion, but they lack a stable association with systematic muscle synergies across movements[36]. In contrast, stepping involves fewer temporal patterns but all structured in stable synergies (a finding we confirmed here), whose fractionation may account for the synergies of older children[36]. Therefore, the spinal locomotor circuits might be reconfigured as a function of sensory feedback from a contact surface.

## Functional significance of developmental changes in complexity and variability of control signals. Low complexity and high variability of neuromuscular signals attest neonatal immaturity but, at the same time, they involve potential benefits for learning locomotor tasks. Not only do infants exhibit few neuromuscular modules, but they also employ a tight synergistic covariation of limb joint angles during stepping[26,32,34,42]. As first suggested by Bernstein[69], when people start learning a skill, they may restrict the number of degrees of freedom to reduce the size of the search space and simplify the coordination problem, lower

dimensionality of control enabling a more efficient exploration of the sensorimotor space. As the skill develops, the novice gradually releases degrees of freedom. The idea of freezing the degrees of freedom of movement at the initial stage of learning a task and then freeing them as learning progresses has been incorporated in neural network models of the development of locomotion[70], as well as in robotic humanoids[71]. It has also been argued that low-dimensional control allows easier generalization across different contexts[5], which is critical to learn different motor tasks during development.

A high variability of the neuromuscular signals has a complementary significance to the low-dimensional character of infant stepping. Motor variability can take two forms: the variability originating from the periphery (sensorimotor noise), which must be decreased to move more accurately, and the variability that originates from central circuits and drives learning-related motor exploration[72]. Indeed, in human adults performing a new task, trial-by-trial variability predicts the individual motor learning ability: the greater the variability, the faster the learning[73,74]. In human infants, the absence of structured variability is a sign of motor disability; the movement stereotypy typical of developmental motor disorders contrasts with the variation and flexibility of healthy infant movements[59].

In conclusion, the nativism/learning controversy for the ontogeny of locomotion that we outlined initially can be resolved by acknowledging that the basic motor components are laid down prior to and at birth, but they are then reconfigured and enriched considerably from extensive experience with continuously changing brain-body-environment interactions.

## Methods

General procedures were similar to those described in ref. 36. All experiments were in accordance with the World Medical Association Declaration of Helsinki for medical research involving human subjects. The Research Ethics Committee of Azienda Sanitaria Locale (Local Health Center) Roma C approved the experiments with the neonates (protocol CEI/15843 study n. 609, and protocol 27593, study n. 38.15). The Research Ethics Committee of Santa Lucia Foundation approved the experiments with a subset of infants and all adults (protocol CE/AG4/PROG.341-01). The Research Ethics Committee of the Veltischev Research and Clinical Institute for Paediatrics of the Pirogov Russian National Research Medical University approved the experiments for another subset of infants (protocol n. 14/18). A parent for the child and all adult participants provided informed written consent to participate in the study after the nature and possible consequences of the study were explained. All children were full-term at birth and had no known pathology. Those who did not step were excluded from the study. Table 1 gives the average characteristics of the eight different groups of recruited participants: full-term neonates (Apgar score ≥8 at 1 and 5 min, uneventful delivery and perinatal history), infants of 6 different age groups, group 1 (g1, age range 4–6 months), group 2 (g2, 6–8 months), group 3 (g3, 8–10 months), group 4 (g4, 10–14 months), toddlers (12–15 months), preschoolers (24–48 months), and adults. These participants were selected from a larger sample because they had full EMG recordings from all 8 tested muscles of at least 7 strides. One neonate (N6) was included for a separate analysis because she performed 5 consecutive strides, but did not have 7 strides (see below). Supplementary Table 1 gives the detailed characteristics of all 79 participants included in the study.

### Experimental protocols

*Neonates.* Neonates were studied in the hospital well-baby maternity ward. Stepping was elicited following established procedures[32,34,36,42]. An experienced pediatrician held the child under the armpits, with the feet soles touching the treadmill surface. Stepping was typically successful when the child was sufficiently aroused. No movement recording was carried out if the child was drowsy or asleep.

Neonates stepped on a treadmill at different speeds. Each experiment started with stepping trials at 0.03 m/s. All these neonates were then tested at 0.05 and 0.1 m/s. Higher speeds (0.15, 0.2 m/s) were tested only if the neonate was able to step at a lower speed. The belt speed was generally changed between the trials (with the child not in contact with the treadmill). Occasionally, the belt speed was changed in mid-trial; in these instances, the steps performed during the acceleration or deceleration phase were not included in the analysis.

*Infants.* Infants were studied in a hospital laboratory or pediatric room. Stepping was elicited in a manner similar to that used for neonates. Twenty-four infants (from groups g1-g4 and toddlers) were tested on a treadmill with incremental

speeds in the range between 0.05 m/s and 0.6 m/s. Twenty-nine infants (from groups g1-g4, toddlers and preschoolers) were tested on a horizontal walkway (Supplementary Table 1). Toddlers were recorded within 4 weeks from the first day of unsupported walking experience (as reported by the parents).

*Adults.* They walked at 1.1 m/s (4 km/h) on a treadmill.

**Experimental setups and recordings.** Neonates, infants (except those who stepped on a horizontal walkway) and adults stepped on a pediatric treadmill model 2 Carlin's Creations (Sturgis, MI, 69 cm × 46 cm length × width), model 3 Carlin's Creations (Sturgis, MI, 81 cm × 46 cm), and standard treadmill (EN-Mill 3446.527, Bonte Zwolle BV, BO Systems, The Netherlands, 150 × 60 cm), respectively.

Kinematics was recorded by means of a video camera (Canon MD160, Canon Inc., Japan, 1152 × 864 pixels, 25 frames/s or Panasonic HC-V760EE-K, 1920 × 1080 pixels, 50 frames/s) for children who stepped on a walkway. We used a 3D SIMI motion capture system (Munich, Germany, 3 video cameras, 640 × 480 pixels, 100 frames/s) to record stepping on treadmill in neonates, and a 3D Vicon Nexus (Oxford, UK, 10 Bonita cameras, 1024 × 1024 pixels, 200 frames/s) in infants who stepped on treadmill and adults. In all cases, adhesive markers (diameter, 9 and 14 mm in neonates and older participants, respectively) were attached to the skin over the hip (greater trochanter, GT), knee (lateral femur epicondyle, LE), ankle (lateral malleolus, LM), and fifth metatarsophalangeal joint (5MT) of the leg facing the video cameras.

In neonates, surface EMG activities were recorded at 2 kHz using the wireless Zerowire system (Aurion Srl, Italy) with miniature (2-mm diameter) surface electrodes (Beckman Instruments), bandwidth of 20–1000 Hz with an overall gain of 1000. To minimize movement artefacts, preamplified EMG sensor units were attached at the experimenter's wrist, and twisted pairs of wires (between electrodes and units) were limited to 25 cm length and fixed along the infant leg using elastic gauze. In infants and adults, EMGs were recorded by means of the Trigno Wireless EMG System (Delsys Inc., Boston, MA, bar electrodes, contact 5 × 1 mm), bandwidth of 20–450 Hz, overall gain of 1000. Sampling rate was 2 kHz for participants stepping on treadmill, and 963 Hz for those stepping on a horizontal walkway. In all participants, we recorded bilaterally from rectus femoris (RF), biceps femoris (BF), tibialis anterior (TA), and gastrocnemius lateralis (LG). In all experiments, sampling of kinematic and EMG data was synchronized. All markers and electrodes used in children were for pediatric use.

**Data analysis.** All data analysis was performed using custom-written programs in Matlab (MathWorks, Natick, MA). The acquired kinematic data were low-pass filtered at 20 Hz with a zero-lag 4th-order Butterworth filter. We selected the sequences with at least three consecutive steps involving alternating (left-right) foot placements. A step was defined as a cyclic movement that included the placement of 5MT ahead of GT. Stance and swing phases were defined on the basis of the timing of the local minima of the vertical position of the foot markers (LM and 5MT). Stride cycle was defined as the time interval between two successive touchdowns by the same foot, and included a step of one foot followed by a step of the contralateral foot.

Raw digitized EMG data were first inspected visually to detect artefacts and remove the corrupted data segment from further analysis. Because of the low skin impedance at each electrode site and the preamplification close to the electrodes, artefacts were infrequent (less than 4% of recorded data were removed). EMG data were high-pass filtered at 60 Hz, full-wave rectified, and low-pass-filtered at 5 Hz (except for the analysis of muscle coherence networks, see section below *Spatiofrequency organization of the muscle activity patterns*) to obtain envelope time-series. All filters were zero-lag 4th-order Butterworth. In order to have a comparable set of data for all participants, the EMG data of seven strides at the same (treadmill) or similar (on ground) speed were used for all analyses, except for the frequency analysis of the cases with five consecutive strides (see section *Intramuscular complexity* in Results). Since the percentage of seven consecutive strides was low (~20%) in most neonates, in order to render all data sets comparable, we randomly sampled seven strides of each individual in the other age groups, making sure that each individual had the reference percentage (20%) of seven consecutive strides of neonates. The processed EMG data were time-interpolated over a normalized 200-point time base. Then, we subtracted the minimum over the cycle from each EMG profile, and normalized the EMG amplitude to the maximum computed over all cycles of a given participant and condition.

*Parametrized frequency analysis of EMG data of individual muscles.* To assess intramuscular activity complexity, the Power Spectral Density (PSD) was calculated for each participant using the Fast Fourier Transform (FFT) algorithm (fft.m function in Matlab) on the time-interpolated EMG data of each muscle, concatenated over the selected 7 strides. The PSD for each muscle was then parametrized using the algorithm proposed by Donoghue et al.[46], which models the PSD as a combination of an aperiodic component and periodic oscillatory peaks. These oscillatory components of the PSD are characterized as frequency regions of power over the aperiodic component, and are modeled as Gaussians. The periodic component $G_n$

is the sum of $N$ total Gaussians, described as:

$$G_n(f) = A * \exp\left(\frac{-(f - f_c)^2}{2\sigma^2}\right) \quad (1)$$

where $A$ is the power of the peak, $f_c$ is the center frequency, $\sigma$ is the standard deviation (bandwidth) of the Gaussian, and $f$ is the frequency vector.

The aperiodic component $L$ is modeled using a Lorentzian function, written as:

$$L(f) = b - \log(k + f^\chi) \quad (2)$$

where $b$ is the broadband offset, $\chi$ is the exponent, and $k$ is the "knee" parameter, accounting for the bend in the aperiodic component. Broadband power refers to fluctuations occurring over a broad range of frequencies. The final outputs of the algorithm are the parameters defining the best fit for the aperiodic component in Eq. 2 and the N Gaussians in Eq. 1. The FOOOF algorithm (version 1.0.1) was used to parameterize EMGs power spectra. Settings for the algorithm were: peak width limits from 0.5/T to 10/T and peak threshold = 2. Power spectra were parameterized across the frequency range from 0.1/T to 10/T. Notice that, in order to deal with the difference in stepping frequency with age (see Table 1), we performed frequency analyses on time-interpolated EMG data. Therefore, the vector of input frequency is not expressed in Hz but in 1/T, where T is the stride duration.

Since the parameters of the aperiodic component are strictly correlated between each other and may be difficult to interpret individually, we introduced two additional parameters that better describe the characteristics of the aperiodic fit. The corrected broadband offset $b* = b - \log(k)$ represents the offset of the aperiodic fit relative to the horizontal slope (overall up/down translation of the whole spectrum), and the corrected knee $k* = k^{(1/\chi)}$ defines the frequency at which the aperiodic fit transitions from horizontal to negative slope. Notice that the exponent reflects the slope of the aperiodic component past the knee inflection point.

In order to measure the degree of EMG irregularity, we calculated for each participant and each muscle the spectral entropy[75] as $SE = -\sum_f p(f) \ln(p(f))$, where $p$ is the relative power (that is the ratio of each PSD value to the total PSD) between 0/T and 10/T. $SE$ was divided by the natural logarithm of the number of PSD samples in order to obtain a value between 0 and 1, where 1 represents the $SE$ of a signal with all frequency components of equal power value (white noise).

*Spatiotemporal organization of the muscle activity patterns.* To assess intermuscular activation complexity, we used a dimensionality-reduction approach. Basic neuromuscular modules were extracted from time-varying profiles of processed EMG activities of all recorded muscles of seven strides for each participant using three different methods: spatial, temporal, and space-by-time decomposition[45]. One neonate (N6) performed five consecutive strides, but did not have the seven strides necessary for the factor analysis mentioned above. For each model, we defined specific measures of consistency of the modules across strides and across participants.

Spatial decomposition model: In this model (Fig. 3a, upper panel), for each stride (s), each muscle pattern (m) is represented as a linear combination of a set of time-invariant (and stride-invariant) weight vectors in the muscles space (muscle synergies, w) activated by a time-varying activation coefficient (basic activation patterns, c), as follows:

$$m^s(t) = \sum_{i=1}^{N} c_i^s(t)w_i + \varepsilon \quad (3)$$

where $t$ is the time, $\varepsilon$ are the residuals, and $N$ is the number of spatial modules. Since the basic activation patterns can vary across strides, the consistency measure for this model was evaluated by calculating, for each basic activation pattern ($c_i$), the average scalar product across all possible pairs of strides, and averaging this value across the $N$ modules (Fig. 3a).

Temporal decomposition model: In this model (Fig. 3a, middle panel), for each stride, each muscle pattern is represented as a linear combination of a set of time-varying basic activation patterns (c, invariant across strides) weighted by a set of muscle synergies (w) that can vary across strides:

$$m^s(t) = \sum_{i=1}^{P} c_i(t)w_i^s + \varepsilon \quad (4)$$

where $P$ is the number of temporal modules. Similarly to the spatial decomposition model, the consistency measure for this model was evaluated by calculating, for each muscle synergy ($w_i$), the average scalar product across all possible pairs of strides, and averaging this value across the $N$ modules (Fig. 3a).

Space-by-time decomposition model: This model (Fig. 3a, lower panel), proposed by Delis et al.[45], assumes that each muscle pattern can be reconstructed starting by a set of muscle synergies (w) and a set of basic activation patterns (c) that do not vary across strides, and which are related by a matrix of activation coefficients ($a_{ij}$)

that can vary across strides:

$$m^s(t) = \sum_{i=1}^{P}\sum_{j=1}^{N} c_i(t) a_{ij}^s w_j + \varepsilon \qquad (5)$$

By assuming $P = N$, we obtain square matrices of activation coefficients. In order to evaluate the consistency of the activation coefficients across strides, for each participant, we first ordered the basic activation patterns in a "chronological" order (with respect to the timing of the main peak) and then we sorted the set of synergies to obtain the resulting activation coefficient matrix as close as possible to a diagonal matrix. The diagonality ($d$) of the activation coefficient matrix was calculated, for each stride, using the following formula:

$$d^s = \frac{\sum_{i=1}^{N} a_{ii}^s}{\sum_{i=1}^{N}\sum_{j=1}^{N} a_{ij}^s} \qquad (6)$$

Factorization algorithms: The principal component analysis (PCA, pca.m function in Matlab) and the NNMF algorithm were used to factorize EMG data that were organized into two different matrices as illustrated in Fig. 3a for spatial and temporal decomposition. PCA was also applied to the EMG data of single strides. The sample-based non-negative matrix tri-factorization (sNM3F) algorithm[45] was used to extract the concurrent spatial and temporal modules and the matrices of activation coefficients from the space-by-time decomposition model. For both NNMF and sNM3F algorithms, the best solution was selected out of 100 runs to avoid local minima of the root-mean-squared residuals. In order to assess the spatial and temporal dimensionality, we applied the factorization methods by varying the number of modules from 1 to 8, and calculated, for each $N$ and each given model, the percent of variance accounted for (VAF), the consistency measures described above, and their slope.

Inter-subject consistency: To assess the consistency of the basic activation patterns and of the muscle synergies across participants of the same group, we first identified for each $N$ the modules that were similar across participants. For all decomposition methods, we sorted the modules of each participant in order to attain the minimum distance (1- cosα) between the basic activation patterns of all participants (in the case of spatial decomposition, the average of the basic activation patterns across strides was used for each module). Since for the space-by-time decomposition there is not a unique relationship between synergies and basic patterns, which are related by the $N \times N$ matrices of activations coefficients, after sorting the basic activation patterns, we sorted also the muscle synergies in order to obtain the resulting activation coefficient matrix as close as possible to a diagonal matrix. We then evaluated the consistency across participants as the average similarity (scalar product) of the basic activation patters or of the synergies of the similar modules between all possible couples of subjects of the same group.

Analysis of residuals: To verify whether there was any systematic structure in the residuals ($\varepsilon$ in Eqs. 3–5) that were not fit by the decomposition models, we computed the similarity (scalar product) between the residuals in all muscles and participants of each age group.

*Simulated data.* To assess the influence of noise on the VAF and on the consistency measures, we simulated different sets of eight rectified EMG data starting from known modules ($N$ equal 2 to 5 modules). Following the method used by Tresch et al.[48], we generated a set of $N$ muscle synergies ($w_i$, Fig. 4a) drawn from exponential distribution with a mean of 10. In order to obtain a set of independent synergies, we ran the random generation 1000 times and chose the set that minimized the scalar product between the $N$ synergies. Mimicking the shape and the characteristics of the basic activation patterns extracted from the experimental data, we generated a set of $N$ basic activation pattern ($c_i$, Fig. 4a) as Gaussians, evenly shifted across the gait cycle, with width inversely proportional to $N$. For each module, the same Gaussian was concatenated unchanged over the 7 strides assumed for the simulation (see the upper panel of Fig. 4a). The data generated from these known $w_i$ and $c_i$ using Eq. 3 were then corrupted by a signal-dependent noise with standard deviation proportional to the noiseless data value by a factor of $\eta$ and low-pass filtered at 5 Hz (as the experimental data). For each $N$ and for each level of noise ($\eta$ ranges from 0.9 to 1.7, corresponding to $r^2$ from 0.89 to 0.62 between noiseless and corrupted EMG data, Fig. 4b), we generated 100 data sets on which we applied the decomposition algorithms (varying the number of modules from 1 to 8), and calculated the VAF and consistency measures. The same analysis was performed on 100 structureless data sets obtained by randomly shuffling all samples of the simulated EMGs, independently for each channel.

Notice that using Eq. 3 with the imposed $w_i$ and $c_i$ (repeated unchanged over the 7 strides) to generate the noiseless simulated EMGs is equivalent to using Eq. 4 with the same stride-invariant $c_i$ and repeating $w_i$ unchanged across strides or using the same stride-invariant $c_i$ and $w_i$ in Eq. 5 assuming that the activation coefficients matrices are identity matrices (Fig. 4a).

In a separate series of simulations, we assessed the effect of cycle-to-cycle variability of either the timing (Supplementary Figure 3a) or the amplitude (Supplementary Figure 3b) of the basic patterns on the consistency measures. In

the former case, we generated EMG sets in which the Gaussians of the 7 strides were independently shifted across the stride cycle by a random interval drawn from a normal distribution with mean 0% and standard deviation 5% of the cycle. In the latter case, we generated EMG sets in which the Gaussians of the 7 strides were independently scaled in amplitude by a random factor drawn from a normal distribution with mean 1 and standard deviation 0.3 of their amplitude. For each $N$ and for each level of noise ($\eta$ ranges from 0.9 to 1.7, corresponding to $r^2$ from ~0.7 to about 0.4 between noiseless and corrupted EMG data), we generated 100 data sets on which we applied the decomposition algorithms (varying the number of modules from 1 to 8), and calculated the VAF and consistency measures.

*Consistency parameters.* For both simulated and experimental data, consistency was computed as the average scalar product (cosα) of the activation patterns or the synergies across all possible pairs of strides. Diagonality was computed as the average ratio of the diagonal activation coefficients to all activation coefficients (Eq. 6). We then considered the maximum value of the slope of change of the consistency measures as a function of the number of modules (from 1 to 8). In a few experimental cases, this maximum value corresponded to 1 module but was poorly defined. This occurred in 13% of all subjects for the spatial decomposition, 6% for the temporal decomposition, and 2% for the space-by-time decomposition. In these cases, we based the identification of the number of modules on both the consistency measures across strides and the VAF. Specifically, when 1 module accounted for <20% of the total data variance, we selected the number of modules corresponding to the second highest value of the consistency slope if this value differed by <0.01 relative to the maximum.

*Cluster analysis of neuromuscular modules.* Muscle synergies and time-varying basic activation patterns were computed according to Eqs. 3 and 4, respectively. Here, however, NNMF was separately applied to each single stride of each participant. We varied the number of synergies or basic patterns from 1 to 8, and for further analyses we retained the smallest number accounting for ≥80% of the variance of EMG profiles. Next, to identify similar synergies or patterns across strides, all $w$ or all $c$ extracted from single strides of all participants of each group and condition were pooled together and partitioned in $k$ mutually exclusive clusters using the $k$-means algorithm[36]. To minimize the possibility of local minima, we performed 100 replications of the algorithm. Because the $k$-means method requires choosing the number of clusters as input, we determined the optimal number of clusters in the range 2 to 20 using the Calinski-Harabasz method[76]. The Calinski-Harabasz index is defined as:

$$CH_k = \frac{SS_B}{SS_W}\frac{(N-k)}{(k-1)} \qquad (7)$$

where $SS_B$ is the overall between-clusters variance, $SS_W$ is the overall within-cluster variance, $k$ is the number of clusters, and $N$ is the number of observations. The overall between-clusters variance $SS_B$ is defined as

$$SS_B = \sum_{i=1}^{k} n_i \|m_i - m\|^2 \qquad (8)$$

where $n_i$ is the number of observations in cluster $i$, $m_i$ is the centroid of cluster $i$, $m$ is the overall mean of the sample data, and $\|m_i - m\|$ is the $L^2$ norm between the two vectors. The overall within-cluster variance $SS_W$ is defined as

$$SS_W = \sum_{i=1}^{k}\sum_{x \in c_i} \|x - m_i\|^2 \qquad (9)$$

where $x$ is a data point, $c_i$ is the i-th cluster, and $\|x - m_i\|$ is the $L^2$ norm between the two vectors.

Well-defined clusters have a large between-clusters variance ($SS_B$) and a small within-cluster variance ($SS_W$). The larger the $CH_k$ ratio, the better the data partition. The optimal number of clusters corresponded to the solution with the highest $CH_k$ value.

The goodness of clusterization for individual muscle synergies and activation patterns was assessed using the silhouette method[36]. The centroid for each cluster was the point with minimum distance from all points in the cluster. As distance measure (in 200-dimensions space), we used [1 – cosα_i], α_i being the angle between points (treated as vectors). The silhouette value is a measure of how similar a given data point is to the other data points in its own cluster, when compared to data points belonging to different clusters. The silhouette $S_i$ for the i-th point is defined as:

$$S_i = \frac{(b_i - a_i)}{\max(a_i, b_i)} \qquad (10)$$

where $a_i$ is the average distance from the $i$-th point to the other points in the same cluster as $i$, and $b_i$ is the minimum average distance from the $i$-th point to points in a different cluster, computed over all clusters. All muscle synergies and all activation patterns with $S \leq 0.2$ were considered unmatched and excluded from the corresponding cluster. For each group of participants, the resulting clusters of activation patterns were ordered chronologically, based on the timing of the main peak relative to the stride cycle.

*Spatiofrequency organization of the muscle activity patterns.* To evaluate the coordination between multiple muscles in the frequency domain, we constructed

the muscle coherence networks of neonates and adults with the method proposed by Boonstra et al.[47]. To this end, the EMGs of the eight muscles were high-pass filtered at 30 Hz (zero-lag 4th-order Butterworth) and rectified using the Hilbert transform. The EMG envelopes from the same seven strides used for the other analyses were then concatenated in time (Fig. 3b, left) in order to compute the inter-muscular coherences. From the processed mean-centered EMG signals, we calculated the magnitude-squared coherence (mscohere.m in Matlab) between all possible couples of muscles ($n = 24$, Fig. 3b).

NNMF was used to decompose this set of inter-muscular coherences (cohe) into frequency components (c) and inter-muscle loadings (w) according to the following model:

$$cohe(f) = \sum_{i=1}^{F} c_i(f)w_i + \varepsilon \qquad (11)$$

where $f$ is the frequency, $\varepsilon$ are the residuals and $F$ is the number of frequency components.

For each frequency component (reflecting the spectral signatures of the given module), it is possible to build an 8x8 matrix from the corresponding inter-muscle loading ($w_i$), since each element of the $w_i$ vector represents the coupling strength between a given pair of muscles. We then constructed, for every subject, a set of $F$ networks, each one characterized by a given spectral content ($c_i$), whose nodes are the 8 muscles and the edges are the elements of the weighted matrix constructed from $w_i$.

To characterize the frequency of each network, we calculated the 50% frequency band as the frequency interval in which the frequency component amplitude exceeds half of its maximum. To evaluate the topography of the networks, we calculated the betweenness-centrality as the fraction of all shortest paths in the network that pass through a given node[47] and the average edge weight across all networks (global), across intra-limb connections, and across interlimb connections.

To investigate the dimensionality (the number of components), we performed NNMF and PCA on inter-muscular coherences by varying the number of modules from 1 to 8, and we calculated the VAF by the reconstructed data.

**Statistics and reproducibility**. Descriptive statistics included means ± SD across all participants of a given group. Sample size for each age group is reported in Table 1. Trials performed by each individual are reported in Supplementary Table 1. The values of $r^2$, VAF, distances between vectors, and diagonality were standardized using the Fisher z-transform (inverse hyperbolic tangent), means and confidence intervals were computed on the transformed values and back-transformed using the inverse transformation. Kolmogorov–Smirnov test was used to assess the null hypothesis that the data come from a normal distribution. The non-parametric Wilcoxon rank sum test was used to evaluate the differences between neonates and adults in the average periodic and aperiodic parameters of PSD and in the average spectral entropy, since the null hypothesis of normality of these data was rejected. The Kruskal–Wallis test was used to evaluate the differences in the average measures between all inter-muscular networks from adults and neonates. Tukey's Honestly Significant Difference test was used for post-hoc comparisons. Two-way ANOVA was used to assess the effect of group, the effect of the number of modules used for the decomposition and their interaction on the average of the z-transformed measures of inter-subject consistency. Reported results were considered significant for $p < 0.05$.

**Reporting summary**. Further information on research design is available in the Nature Portfolio Reporting Summary linked to this article.

## Data availability

All data generated or analyzed during this study are included in this published article (and its supplementary information files) and in Supplementary Data 1. Additional data are available from the corresponding author upon reasonable request.

## Code availability

All analysis codes are published and appropriate references are given in the article. Implementation details will be provided from the corresponding author upon reasonable request.

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

## Acknowledgements

We thank Tjeerd Boonstra and Andrea d'Avella for their helpful comments on the muscle network and NNMF analysis, respectively. This work was supported by the Italian Ministry of Health (Ricerca corrente, IRCCS Fondazione Santa Lucia, Ricerca Finalizzata RF-2019-12370232), the Italian Space Agency (grant I/006/06/0 and grant 2019-11-U.0), the Italian University Ministry (PRIN grant 2017CBF8NJ_005 and 2020EM9A8X_003), and INAIL BRIC 2019.

## Author contributions

F.S.L., Y.I., and F.L. designed the study. F.S.L., V.L.S., G.C., A.D., A.F., I.A.S., V.M., Y.I., and F.L. performed experiments. F.S.L., Y.I., and F.L. analyzed data. G.C., A.F., I.A.S., and V.M. provided key populations. F.S.L., Y.I., and F.L. wrote the paper with input from all authors.

## Competing interests

The authors declare no competing interests.
