## [Peer Review File · Communications Biology]

Reviewers' comments:

Reviewer #1 (Remarks to the Author):

Manuscript Submitted to Communications Biology

ID: COMMSBIO-22-1956-T

Title: Complexity and variability of modular neuromuscular control during human locomotor development.

By Drs. Francesca Sylos-Labini, Valentina La Scaleia, Germana Cappellini, Arthur Dewolf, Adele Fabiano, Irina A. Solopova, Vito Mondì, Yury Ivanenko, Francesco Lacquaniti.

General comments:

At the beginning of the abstract, the authors raised a question "When does modular control of locomotion emerge during human development?", followed by two contrasting views. One view is that modularity is not innate, being learnt over several months of experience. Alternatively, the basic motor modules are present at birth, but are subsequently reconfigured due to changing brain-body environment interactions. I think the question is getting obsolete, considering surprising repertoires of so-called "general movements" in utero. On the other hand, this study collected extremely precious muscle activities during stepping movement from neonates, infants, toddlers, preschool children and adults. There is no doubt that this dataset is unprecedented in this area. Furthermore, the authors performed sophisticated linear decomposition analyses on the EMG data to improve reliability of the decomposition and identified the modularity even in neonates. Nevertheless, the authors appear to focus only on the number of modules and pay much less attention in description of functional comparison of the modularity between different age groups. In other words, the current version of this paper is largely limited to establishment of the decomposition methods. I believe that general readers will highly appreciate more functional characterization of identified modules, in particular comparisons between different age groups within the dataset.

Specific comments and suggestions

1. It is important to discuss why you selected the eight muscles (four in each leg) in this and previous experiments. In other words, how it is optimal in analysis of modularity in stepping or locomotion?
2. Line 130, the broadband offset -> I would appreciate more explanation.
3. Lines 213-216, "identified 2.08 ± 0.62 (mean \pm SD), 1.89 ± 0.74 , 1.77 ± 0.58 , 2.48 ± 0.91 , 2.67 ± 1.25 , 2.55 ± 1.13 , 3.25 ± 1.20 , and 5.4 ± 1.2 modules in neonates, 4-6, 6-8, 8-10, 10-12 months old infants, toddlers, preschoolers, and adults, respectively (Fig. 5c)" -> identified 2.08 ± 0.62 (mean \pm SD)(neonates), 1.89 ± 0.74 (4-6m infants), 1.77 ± 0.58 (6-8m infants), 2.48 ± 0.91 (8-10m infants), 2.67 ± 1.25 (10-12m infants), 2.55 ± 1.13 (toddlers), 3.25 ± 1.20 (preschoolers), and 5.4 ± 1.2 (adults) modules in each age group (Fig. 5c)
4. Lines 216-219, "The trend with age was statistically significant: the linear regression of the number of modules versus age (using all individual values of the identified number of modules and individual age expressed in months) yielded $r=0.86$ including only all children, and $r=0.94$ including also the adults." It would be helpful to add a figure for this sentence, in particular for the latter half.
5. Line 239, "this is the case for 2 modules but not 4 modules" -> this is the case for 2 modules but not for 4 modules
6. Line 304, "(c 1-c 2 in Fig. 8a and c 1-c in Fig. 9b)" -> (c 1-c 2 in Fig. 9a and c 1-c in Fig. 9b)
7. Lines 681-695, the silhouette method -> It would be very helpful if you add a supplementary figure for explanation.

Reviewer #2 (Remarks to the Author):

In this MS, the authors address a very important but difficult question in developmental motor neuroscience: How does the complexity of locomotor outputs evolve across developmental stages in humans? This question is highly relevant to understanding how innate and learned factors contribute to the shaping of mature motor commands. Complexity is defined and quantified here as the number of motor modules (i.e., the dimensionality) identifiable from multi-muscle

electromyographic signals (EMGs) recorded during locomotion. Determining the correct dimensionality from EMGs of neonates and infants remains a difficult problem because such data are often noisy. The authors attempted to overcome this challenge by a novel method of dimensionality identification that relies on the consistency of factorization outputs across consecutive strides, and by additional muscle coherence analysis in the spatio-frequency domain. With these methods, the authors found that across the developmental stages from neonates to adults, EMG complexity increases while the overall EMG variability decreases. Thus, locomotor maturity is characterized by more stable recruitment of a higher number of motor modules.

Overall, this is an interesting, thought-provoking work that addresses a controversial topic. The authors are commended for their analysis of data from a large number of subjects at different developmental stages, their use of multiple different models of motor modularity, and their consideration of methods beyond the popular factorization algorithms. Their newly proposed method of finding dimensionality will also be a significant contribution to the field. While their conclusion of complexity increase is reasonably compelling, its validity rests critically on their new method of dimensionality determination by stride consistency assessment, whose reliability in turn depends on certain assumptions on data structure which must be further clarified and examined. I summarize below what the authors should attend to in their revision so that this work can hopefully be even more impactful.

MAJOR COMMENTS

(1) The authors' method for finding the number of motor modules relies on assessing either the consistency of factorization outputs (for spatial or temporal decomposition) or the diagonality of the activation coefficients (for space-by-time decomposition) across the number of modules, and then finding the number at which there was the largest drop in consistency or diagonality (Fig. 4). The validity of this method was verified on simulated EMGs corrupted by signal-dependent noise. The important assumption here is that the underlying basic patterns (for spatial decomposition), synergies (for temporal), or coefficient diagonality (for space-by-time) that generate the EMGs are constant through the strides examined. In fact, the simulated EMGs were generated with this assumption of constancy (line 643, "for each module, the same [simulated] Gaussian was concatenated unchanged over the 7 strides assumed for the simulation"; also graphically depicted in Fig. 4A). But I wonder how the reliability of this method may change when the underlying basic patterns/synergies/coefficient diagonality naturally varies from cycle to cycle (i.e., they are naturally inconsistent). In this case, both their natural inconsistencies and over-fitting of modules would contribute to the decrease in consistency/diagonality as the number of modules increases, thus making it difficult to find the "correct" number of modules.

For example, in Fig. 5C, in one extreme interpretation, one may argue that across the 8 time points the real number of underlying modules could be the same, but then during development, the underlying generative structures themselves become more and more consistent in their recruitment across cycles, thus driving the data structure closer to the consistency assumption required by the method. The apparent low dimensionality of the earlier stages is then the consequence of the natural cycle-to-cycle variability of the basic pattern/synergies/coefficient diagonality that drives the consistency measure down at a small number of modules. In fact, in the recent preprint by Hinnekens, Teulier et al. (2022), their argument is exactly that the coefficient diagonality increases with age.

I suppose the authors could further examine the reliability of their method in simulated data by relaxing the above assumption. Can the correct number of modules still be identified when the timing or amplitude of the basic patterns/synergies/coefficient diagonality are themselves corrupted by noise?

A related suggestion is that since diagonality was calculated for each stride individually (equation 6), it appears that in Fig. 4B it was the average diagonality of the 7 strides that was used for dimensionality determination (this is not clearly mentioned in the MS). But to reflect consistency of diagonality across strides, would the variance of diagonality across strides convey even more information?

(2) Related to (1), in the MS there is not any clear description of the rationale behind using consistency measures for determining the correct number of modules. Specifically, in lines 155-165, while the limitations of methods based on VAF thresholds are elaborated at length, there is no explanation on what the consistency measures are, and why they "make sense". Since the use of these measures for dimensionality determination is new in the field and the result from this analysis is the core argument of this paper, the authors should make an effort to explain these measures better and more clearly.

(3) The Introduction can provide better framing of this study. After a presentation of the 3 hypotheses on motor module development (learning hypothesis/nativist view/combined roles of innate and learned components), the authors immediately jumped to a discussion on the difficulty in correctly determining the number of motor modules in EMGs from infants. The authors should elaborate on how these 3 hypotheses may be at least partially resolved experimentally and/or how measures of complexity and variability may contribute to this resolution, and then describe in broad terms the unique approaches adopted here for quantifying EMG complexity and variability across developmental stages.

MINOR COMMENTS

(1) The phrase "high variability" should be used with more precision. In some instances, they appear to mean high levels of noise (e.g., line 6 in abstract, lines 68-69 etc.), while in other instances, the same phrase is used to mean high levels of structured EMG variability (e.g., line 11, "low complexity and high variability of neuromuscular signals"). I have found this lack of clarity to be a very major source of confusion, especially since the presence of structured EMG variability should aid the identification of motor modules, in principle.

(2) Fig. 5: I would suggest reporting the number of modules identified using traditional VAF-threshold based measures (from PCA or NNMF) as well and assessing whether these numbers correlate with the numbers from consistency-based measures. Judging from Fig. 5A, at least for the neonates, infants and toddlers/preschoolers groups, they appear to correlate.

(3) Line 540: "each individual had the reference percentage of neonates" – it is unclear what the reference percentage of neonates refers to.

(4) Line 546: Insert (PSD) after Power Spectral Density.

(5) Line 552: It is unclear whether the "peaks" refer to the periodic or aperiodic components.

Reviewer #3 (Remarks to the Author):

The study investigates whether variability in EMG signals during walking at different life stages affects the ability to identify modularity in muscle control during locomotion. The authors use EMG recordings as well as simulated data to suggest that there is increased modularity as humans develop and increased complexity. Their analysis also suggests differences in the organization of intralimb and interlimb coordination between earlier and later stages of development.

Changes in modular control of muscles during locomotor development is highly relevant to understanding motor control. The analysis in the study is done methodically, and the conclusions are generally supported by the data. Some of the conclusions about increased modularity with development have been demonstrated previously by many studies, including the authors' own work. The paper's novelty lies in the analysis of how variability can influence the estimation of modularity and the use of different methods of factorization to analyze the EMG data. The analysis of intramuscular complexity is also novel.

The paper's readability for readers that are less familiar with the field could be improved by providing more details to facilitate comprehension of certain technical aspects.

Issues:

Line 72. Not sure what "variable amounts of noise concur to the overall output" means. The term "concur" does not seem to fit here

Line 74. Due to the difficulty of determining the number of basis vectors? Clarify what it is that is difficult?

Line 82. The authors stated earlier that the identification of modules has only been done with factorization but no alternatives are proposed. Instead, the study uses three different factorization methods. Please clarify why factorization was still used and how the three different factorization methods differ and whether some of these methods could be an improvement or a departure over previous studies?

Line 130-131, please describe qualitatively what these parameters of the aperiodic component mean.

Line 142-152. The rationale for choosing 7 strides is not evident and should be explained.

Line 171. Not sure what $N=2/5$ modules means

Lines 172-174. Additional insights into the possible values of these three methods of EMG factorization would be valuable

Lines 217. I suggest adding a graph of number of modules versus age in the figure or in a supplemental figure as well as regression lines

Lines 220. Please summarize section of lines 220-226 in terms of its meaning.

Lines 238-239. The conclusion is not obvious at all from the figure 6. Please consider referencing to exactly what the matrix of the activation coefficients are on the figure

Please consider citing some recent work suggesting changes in postnatal maturation of sensorimotor integration in the spinal cord (for example, see Laliberte et al Frontiers Neural Circuits 2022) as a mechanism for changes in locomotor control during development

Line 492. The neonates and infants stepped on the treadmill at different speeds while the adults stepped at one speed only (line 492). Would this affect the amount of variability seen at the different stages?

Figures:

I would suggest a redesign of Figure 3a. Unfortunately, it does not convey an easily interpretable sense of the differences between the three techniques

Figure 4 and 7. What is $\cos \alpha$ as a measure of consistency? The explanation in the Methods does not provide sufficient details to fully understand

Figure 5. What is the difference between patterns consistency, synergies consistency and diagonality? Why are there so many individuals with only 1 module even in preschoolers? What does this mean?

The manuscript was reviewed by Tuan Bui, University of Ottawa

Reviewer #1	Authors' response
General comments: At the beginning of the abstract, the authors raised a question “When does modular control of locomotion emerge during human development?”, followed by two contrasting views. One view is that modularity is not innate, being learnt over several months of experience. Alternatively, the basic motor modules are present at birth, but are subsequently reconfigured due to changing brain-body environment interactions. I think the question is getting obsolete, considering surprising repertoires of so-called “general movements” in utero.	Following the reviewer’s comment, we incorporated a better justification for this question in the revised Introduction. At line 35 we added: “Although these animals show a rich repertoire of primitive movements (such as general movements, kicking or stepping in human babies) well before birth which persist over a variable time after birth, the structure of the motor commands underlying these movements is still under dispute. Moreover, how innate and learned factors contribute to the progressive shaping of motor commands remains poorly understood.” Our work certainly goes along with the reviewer’s viewpoint that modularity is present early on in motor development, but the references in the ms (plus many others not included for brevity) show that the issue of innateness versus learned origin of modularity of motor control is far from being resolved and remains highly controversial in developmental motor science.
On the other hand, this study collected extremely precious muscle activities during stepping movement from neonates, infants, toddlers, preschool children and adults. There is no doubt that this dataset is unprecedented in this area. Furthermore, the authors performed sophisticated linear decomposition analyses on the EMG data to improve reliability of the decomposition and identified the modularity even in neonates.	We thank the reviewer for the appreciation of our extensive work.
Nevertheless, the authors appear to focus only on the number of modules and pay much less attention in description of functional comparison of the modularity between different age groups. In other words, the current version of this paper is largely limited to establishment of the decomposition methods. I believe that general readers will highly appreciate more functional characterization of identified modules, in particular comparisons between different age groups within the dataset.	Following the reviewer’s recommendation, we added a new section starting from line 324: “FUNCTIONAL COMPARISON OF THE MODULARITY BETWEEN NEONATES AND ADULTS. Figure 6a and Figure 5Sa show that the 2 main patterns of EMG activity in neonates exhibit sinusoidal-like waveforms, the first pattern (c1) peaking at ~30% of the cycle (midstance), and the second one (c2) peaking at ~75% (midswing). On each limb, c1 recruits the quasi-synchronous activation of several extensor and flexor muscles, contributing to stiffen the limb and to exert vertical forces supporting part of body weight^{31,33,35,41}. Neonates typically support ~30-40% of their weight during this stepping

	phase⁴². c2 recruits mainly flexor muscles such as tibialis anterior, contributing to flex the leg and foot. Importantly, neonates lack major muscle activity at either touch-down or lift-off, and they show correspondingly small tangential forces at the step-by-step transitions³¹. Forward progression/propulsion is provided by the experimenter (or treadmill belt) rather than by the neonate. Overall, the sequence of muscle activations generates the idiosyncratic style of locomotion of neonates, involving mainly digitigrade foot contact and hyperflexion of the lower limbs during swing^{31,33}. Figure 6c and Figure 5Sc show that adults exhibit 4 main patterns of EMG activity, each of much shorter duration relative to the neonatal patterns. On each limb, these patterns are accurately timed around the four critical events of the gait cycle, heel strike, weight acceptance/forward propulsion, lift-off, and touch down. Adults show a much lower extent of muscle co-contraction as compared with neonates and infants^{41,42}. During stance, the limbs are kept relatively extended, and the center of pressure on the ground shifts smoothly heel-to-toe.”
1. It is important to discuss why you selected the eight muscles (four in each leg) in this and previous experiments. In other words, how it is optimal in analysis of modularity in stepping or locomotion?	Starting from line 443, we now specify: “In this study, we recorded the EMG activity of eight muscles, rectus femoris, biceps femoris, tibialis anterior, and gastrocnemius lateralis of both lower limbs. These muscles were selected because they correspond to those analysed in several previous studies in neonates and infants^{26,31,32,36,41}. While a larger set of muscles would allow a more detailed description of neuromuscular control of locomotion, in a previous study³⁵ we showed that the basic activation patterns did not differ appreciably when extracted from the present set of muscles (n = 8) or a larger set of muscles (n = 22). Therefore, we believe that the main conclusions of our work would not change by including a larger set of muscles.”
2. Line 130, the broadband offset -> I would appreciate more explanation	At line 644 and from line 653, we now explain the meaning of the different parameters.
3. Lines 213-216, “identified 2.08 ± 0.62 (mean \pm SD), 1.89 ± 0.74, 1.77 ± 0.58, 2.48 ± 0.91, 2.67 ± 1.25, 2.55 ± 1.13, 3.25 ± 1.20, and 5.4 ± 1.2 modules in neonates, 4-6, 6-8, 8-10, 10-12 months old infants, toddlers, preschoolers, and adults, respectively (Fig. 5c)” -> identified 2.08	We modified this section starting from line 257 and adding a new panel to Figure 5: “Overall, the consistency measures showed a trend toward increasing dimensionality of the modules with increasing age. Figure 5d plots the mean values of the modules identified

± 0.62 (mean \pm SD)(neonates), 1.89 ± 0.74 (4-6m infants), 1.77 ± 0.58 (6-8m infants), 2.48 ± 0.91 (8-10m infants), 2.67 ± 1.25 (10-12m infants), 2.55 ± 1.13 (toddlers), 3.25 ± 1.20 (preschoolers), and 5.4 ± 1.2 (adults) modules in each age group (Fig. 5c)	using the maximum of the slope of diagonality of the activation coefficients matrix (space-by-time decomposition) versus the mean age of each group of participants. The trend with age was statistically significant: the linear regression of the number of modules versus age yielded $r=0.88$ including only all children, and $r=0.95$ including also the adults. Very similar results were obtained using the number of modules identified from the consistency measures for the spatial and temporal decomposition."
4. Lines 216-219, "The trend with age was statistically significant: the linear regression of the number of modules versus age (using all individual values of the identified number of modules and individual age expressed in months) yielded $r=0.86$ including only all children, and $r=0.94$ including also the adults)." It would be helpful to add a figure for this sentence, in particular for the latter half.	As explained in the previous response, we added a new panel to the Figure as suggested by the reviewer.
5. Line 239, "this is the case for 2 modules but not 4 modules" -> this is the case for 2 modules but not for 4 modules	Done (at line 290-291).
6. Line 304, "(c 1-c 2in Fig. 8a and c 1-c in Fig. 9b)" -> (c 1-c 2in Fig. 9a and c 1-c in Fig. 9b)	Thank you for picking up this mistake, which we now corrected at line 380.
7. Lines 681-695, the silhouette method -> It would be very helpful if you add a supplementary figure for explanation.	We now added a new Supplementary Figure (Supplementary Figure 6) to document the results of the cluster analysis and the silhouette method.
Reviewer #2	Authors' response
In this MS, the authors address a very important but difficult question in developmental motor neuroscience: How does the complexity of locomotor outputs evolve across developmental stages in humans? This question is highly relevant to understanding how innate and learned factors contribute to the shaping of mature motor commands. Complexity is defined and quantified here as the number of motor modules (i.e., the dimensionality) identifiable from multi-muscle electromyographic signals (EMGs) recorded during locomotion. Determining the correct dimensionality from EMGs of neonates and infants remains a difficult problem because such data are often noisy. The authors attempted to overcome this challenge by a novel method of dimensionality identification that relies on the consistency of factorization	We thank the reviewer for the appreciation of our extensive work.

outputs across consecutive strides, and by additional muscle coherence analysis in the spatio-frequency domain. With these methods, the authors found that across the developmental stages from neonates to adults, EMG complexity increases while the overall EMG variability decreases. Thus, locomotor maturity is characterized by more stable recruitment of a higher number of motor modules. Overall, this is an interesting, thought-provoking work that addresses a controversial topic. The authors are commended for their analysis of data from a large number of subjects at different developmental stages, their use of multiple different models of motor modularity, and their consideration of methods beyond the popular factorization algorithms. Their newly proposed method of finding dimensionality will also be a significant contribution to the field. While their conclusion of complexity increase is reasonably compelling, its validity rests critically on their new method of dimensionality determination by stride consistency assessment, whose reliability in turn depends on certain assumptions on data structure which must be further clarified and examined. I summarize below what the authors should attend to in their revision so that this work can hopefully be even more impactful.	
MAJOR COMMENTS (1) The authors' method for finding the number of motor modules relies on assessing either the consistency of factorization outputs (for spatial or temporal decomposition) or the diagonality of the activation coefficients (for space-by-time decomposition) across the number of modules, and then finding the number at which there was the largest drop in consistency or diagonality (Fig. 4). The validity of this method was verified on simulated EMGs corrupted by signal-dependent noise. The important assumption here is that the underlying basic patterns (for spatial decomposition), synergies (for temporal), or coefficient diagonality (for space-by-time) that generate the EMGs are constant through the strides examined. In fact, the simulated EMGs were generated with this assumption of constancy (line 643, "for each module, the same [simulated] Gaussian was concatenated unchanged over the 7 strides	Following the reviewer's suggestion we carried out new simulations which confirmed the previous results. We added new text and a new supplementary Figure (Supplementary Fig. 3) reporting these results. At line 229, we added: "In the simulations of Fig. 4 and Supplementary Figure 2, the underlying basic patterns (for spatial decomposition), synergies (for temporal decomposition), or activation coefficients (for space-by-time decomposition) that generated the EMGs were constant across the strides before being corrupted by different levels of noise. In a different set of simulations, we relaxed this constraint and assessed the effect of cycle-to-cycle variability of either the timing (Supplementary Figure 3a) or the amplitude (Supplementary Figure 3b) of the basic patterns on the consistency measures. In Fig. 3S, the r^2 between noiseless and noisy EMG data ranged

assumed for the simulation”; also graphically depicted in Fig. 4A). But I wonder how the reliability of this method may change when the underlying basic patterns/synergies/coefficient diagonality naturally varies from cycle to cycle (i.e., they are naturally inconsistent). In this case, both their natural inconsistencies and over-fitting of modules would contribute to the decrease in consistency/diagonality as the number of modules increases, thus making it difficult to find the “correct” number of modules. For example, in Fig. 5C, in one extreme interpretation, one may argue that across the 8 time points the real number of underlying modules could be the same, but then during development, the underlying generative structures themselves become more and more consistent in their recruitment across cycles, thus driving the data structure closer to the consistency assumption required by the method. The apparent low dimensionality of the earlier stages is then the consequence of the natural cycle-to-cycle variability of the basic pattern/synergies/coefficient diagonality that drives the consistency measure down at a small number of modules. In fact, in the recent preprint by Hinnekens, Teulier et al. (2022), their argument is exactly that the coefficient diagonality increases with age. I suppose the authors could further examine the reliability of their method in simulated data by relaxing the above assumption. Can the correct number of modules still be identified when the timing or amplitude of the basic patterns/synergies/coefficient diagonality are themselves corrupted by noise?	between about 0.7 and 0.4 for η ranging between 0.9 and 1.7. We found that, also in this case, the slope of the consistency measures exhibited a peak (especially marked for the diagonality of the space-by-time decomposition) corresponding to the actual number of simulated modules ($N=3$), which remained stable over a wide range of noise levels.” The corresponding Methods section was also revised to describe the new simulations.
A related suggestion is that since diagonality was calculated for each stride individually (equation 6), it appears that in Fig. 4B it was the average diagonality of the 7 strides that was used for dimensionality determination (this is not clearly mentioned in the MS). But to reflect consistency of diagonality across strides, would the variance of diagonality across strides convey even more information?	We looked into this parameter but we did not find anything consistent worth of being reported. The reason is that the variance of both the diagonality and its slope are very low and do not discriminate.
(2) Related to (1), in the MS there is not any clear description of the rationale behind using consistency measures for determining the correct number of modules. Specifically, in lines 155-165, while the limitations of methods	We revised the text at several points to take this suggestion into account. At lines 93-94 we specify:

based on VAF thresholds are elaborated at length, there is no explanation on what the consistency measures are, and why they “make sense”. Since the use of these measures for dimensionality determination is new in the field and the result from this analysis is the core argument of this paper, the authors should make an effort to explain these measures better and more clearly.	“Dimensionality was quantified by means of both VAF and consistency measures well suited to take the presence of noise into account.” At line 215 and following, we added: “Consistency was computed as the average scalar product ($\cos\alpha$) of the basic patterns or the synergies across all possible pairs of strides. These measures estimate the similarity of a given pattern or synergy across strides. Diagonality was computed as the average ratio of the diagonal activation coefficients to all activation coefficients (Eq. 6 in Methods). Activation coefficients in the space-by-time decomposition represent the level of activation of each possible pair of spatial and temporal modules⁴⁴. We then considered how these consistency measures change as a function of the number of modules (from 1 to 8), by taking the slope of this function (lower panels in Fig. 4b). Maximum slope locates the point of the function where increasing or decreasing the number of modules relative to this point leads to the most drastic change of the corresponding consistency parameter. Thus, the slope represents a sensitive indicator for the dimensionality of modules in the presence of noise.”
(3) The Introduction can provide better framing of this study. After a presentation of the 3 hypotheses on motor module development (learning hypothesis/nativist view/combined roles of innate and learned components), the authors immediately jumped to a discussion on the difficulty in correctly determining the number of motor modules in EMGs from infants. The authors should elaborate on how these 3 hypotheses may be at least partially resolved experimentally and/or how measures of complexity and variability may contribute to this resolution, and then describe in broad terms the unique approaches adopted here for quantifying EMG complexity and variability across developmental stages.	Following the reviewer’s suggestion, we added at line 71: “The different hypotheses outlined above may be resolved experimentally by carefully comparing the EMG activities at several different stages of motor development. Moreover, measures of complexity and variability may contribute to the resolution.” At line 93 we now specified: “Dimensionality was quantified by means of both VAF and consistency measures well suited to take the presence of noise into account. Consistency was estimated as a similarity index of the synergies across all strides for the spatial decomposition, a similarity index of the temporal patterns across all strides for the temporal decomposition, and as a diagonality index of the activation coefficients across all strides for the space-by-time decomposition. We found that the rate of change of these consistency measures as a function of the hypothetical number of modules was a

	sensitive indicator of the dimensionality of the modules in the presence of noise.”
MINOR COMMENTS (1) The phrase “high variability” should be used with more precision. In some instances, they appear to mean high levels of noise (e.g., line 6 in abstract, lines 68-69 etc.), while in other instances, the same phrase is used to mean high levels of structured EMG variability (e.g., line 11, “low complexity and high variability of neuromuscular signals”). I have found this lack of clarity to be a very major source of confusion, especially since the presence of structured EMG variability should aid the identification of motor modules, in principle.	We revised the text at several places to take this comment into account. At line 6 we now specify “presence of noise” At line 321 we specify “Therefore, in neonates all EMG activity not accounted for by 2 spatio-temporal modules reflects unstructured step-by-step variability (noise).” At line 410 we specify “we tackled the potential confound due to noise and structured variability by means of different approaches” At line 531 we specify “In human infants, the absence of structured variability is a sign of motor disability”
(2) Fig. 5: I would suggest reporting the number of modules identified using traditional VAF-threshold based measures (from PCA or NNMF) as well and assessing whether these numbers correlate with the numbers from consistency-based measures. Judging from Fig. 5A, at least for the neonates, infants and toddlers/preschoolers groups, they appear to correlate.	The observation of the reviewer is correct. Indeed, there is a qualitative trend also using VAF thresholds. However, as shown with the simulations in the previous section (Fig. 4 and Supplementary Figure 2), using a VAF threshold is not sufficient to estimate unambiguously the dimensionality of muscle activation modules in datasets affected by noise. Accordingly, although a trend can be revealed with VAF in our experimental data, one cannot expect a simple correlation between the number of modules detected by traditional VAF-thresholds and the number of modules detected by the consistency measures, which are the most reliable in the presence of noise as shown by the simulations. At line 245 we specify “For a number of modules less than 8 (the maximum number that is theoretically possible, given that there are 8 muscles), VAF was higher in neonates and decreased with age in infants (a trend especially clear with the spatial decomposition model, Fig. 5a), consistent with an increasing dimensionality with age. However, the differences across age groups were small and insufficient to assess the dimensionality of the data sets unambiguously.” To take the reviewer’s suggestion into account, at line 272 we added “Critically, however, the number of modules necessary to account for a

	given level of variance tended to increase with age, consistent with the hypothesis that dimensionality of neuromuscular control increases with age.”
(3) Line 540: “each individual had the reference percentage of neonates” – it is unclear what the reference percentage of neonates refers to	At line 627 we now specify “making sure that each individual had the reference percentage (20%) of 7 consecutive strides of neonates”
(4) Line 546: Insert (PSD) after Power Spectral Density	Done (at line 633)
(5) Line 552: It is unclear whether the “peaks” refer to the periodic or aperiodic components	We revised the text at lines 637 and following. “PSD as a combination of an aperiodic component and periodic oscillatory peaks.”
Reviewer #3	Authors’ response
The study investigates whether variability in EMG signals during walking at different life stages affects the ability to identify modularity in muscle control during locomotion. The authors use EMG recordings as well as simulated data to suggest that there is increased modularity as humans develop and increased complexity. Their analysis also suggests differences in the organization of intralimb and interlimb coordination between earlier and later stages of development. Changes in modular control of muscles during locomotor development is highly relevant to understanding motor control. The analysis in the study is done methodically, and the conclusions are generally supported by the data. Some of the conclusions about increased modularity with development have been demonstrated previously by many studies, including the authors' own work. The paper's novelty lies in the analysis of how variability can influence the estimation of modularity and the use of different methods of factorization to analyze the EMG data. The analysis of intramuscular complexity is also novel. The paper's readability for readers that are less familiar with the field could be improved by providing more details to facilitate comprehension of certain technical aspects.	We thank the reviewer for the appreciation of our extensive work.
Issues: Line 72. Not sure what "variable amounts of noise concur to the overall output" means. The term "concur" does not seem to fit here	We revised the text at line 79 and following: “variable amounts of noise affect the overall output”
Line 74. Due to the difficulty of determining the number of basis vectors? Clarify what it is that is difficult?	At line 80 we revised the text by deleting the part on the difficulty:

	“In fact, a systematic method to determine the number of the basis vectors has not been established by now.”
Line 82. The authors stated earlier that the identification of modules has only been done with factorization but no alternatives are proposed. Instead, the study uses three different factorization methods. Please clarify why factorization was still used and how the three different factorization methods differ and whether some of these methods could be an improvement or a departure over previous studies?	Following the reviewer’s suggestion, we deleted the previous sentence “Third, in several previous developmental studies, the dimensionality of neuromuscular control was addressed only using factorization methods, such as non-negative matrix factorization or independent component analysis”. Moreover, we now added at line 86: “We used 3 different methods for EMG factorization, spatial decomposition, temporal decomposition, and space-by-time decomposition⁴⁴. Each of these methods makes specific assumptions. Thus, the spatial decomposition assumes spatial modularity, the temporal decomposition assumes temporal modularity, and the space-by-time decomposition assumes the concurrent existence of spatial and temporal modules⁴⁴”
Line 130-131, please describe qualitatively what these parameters of the aperiodic component mean.	Starting from line 147 we describe qualitatively the significance of the parameters and then provide full details in Methods (starting from line 643).
Line 142-152. The rationale for choosing 7 strides is not evident and should be explained.	Starting from line 118 we expanded the text to provide the rationale: “The constraint of 7 strides arose from the results obtained in neonates, who typically perform a limited number of steps^{26,31,35,36}. Here, we found that 7 strides was the maximum number of strides that was common to all neonates.”
Line 171. Not sure what N=2/5 modules means	At line 191 we specified more clearly: “number of simulated modules ranging from 2 to 5 ”
Lines 172-174. Additional insights into the possible values of these three methods of EMG factorization would be valuable	Following the reviewer’s suggestion, at line 87 we added “Each of these methods makes specific assumptions. Thus, the spatial decomposition assumes spatial modularity, the temporal decomposition assumes temporal modularity, and the space-by-time decomposition assumes the concurrent existence of spatial and temporal modules.”
Lines 217. I suggest adding a graph of number of modules versus age in the figure or in a supplemental figure as well as regression lines	We now added a new panel Figure (5d) plotting the number of modules versus age with regression lines.
Lines 220. Please summarize section of lines 220-226 in terms of its meaning	At line 265 we now specify “Notice that the VAF by two modules in neonates was significantly lower than the VAF by four modules in adults for all decomposition models, due to higher noise (unstructured variability) in the former than the latter.”

Lines 238-239. The conclusion is not obvious at all from the figure 6. Please consider referencing to exactly what the matrix of the activation coefficients are on the figure	Starting from line 283, we now specify in detail “The same data are analyzed using the space-by-time decomposition in Fig. 6. This method, in particular, allows a clear visual assessment of how changing the number of dimensions would affect the way one can account for the data in the presence of variability⁴⁴. In particular, the matrix of the activation coefficients for all single strides indicates the extent to which the participants of a given age group use the same modules (patterns and synergies) in all strides. Thus, higher values of the activation coefficients along the matrix diagonal (a_{ii}) relative to the activation coefficients off-diagonal (a_{ij} and a_{ji}) denote a greater consistency of engagement of the same modules in most strides across participants of a given age group. Figure 6 shows that in neonates this is the case for 2 modules but not for 4 modules, consistent with our previous quantitative assessment. Indeed, it can be noticed that the coefficients a_{11} and a_{22} are much higher than a_{12} and a_{21} for 2 modules in neonates (Fig. 6a), whereas a_{11}, a_{22}, a_{33} and a_{44} are only slightly higher than the other coefficients for 4 modules (Fig. 6b). By contrast, a_{11}, a_{22}, a_{33} and a_{44} are much higher than the other coefficients for 4 modules in the adults (Fig. 6c).”
Please consider citing some recent work suggesting changes in postnatal maturation of sensorimotor integration in the spinal cord (for example, see Laliberte et al Frontiers Neural Circuits 2022) as a mechanism for changes in locomotor control during development	At line 457 we now cite this interesting article (Although it deals with postnatal maturation of grasping rather than locomotion, the described mechanism may well apply to locomotion as well.)
Line 492. The neonates and infants stepped on the treadmill at different speeds while the adults stepped at one speed only (line 492). Would this affect the amount of variability seen at the different stages?	At line 119 we added: “In order to have a comparable set of data for all participants, the EMG data of 7 strides at the same (treadmill) or similar (on ground) speed were used for all analyses.”
Figures: I would suggest a redesign of Figure 3a. Unfortunately, it does not convey an easily interpretable sense of the differences between the three techniques	We modified Fig. 3a by adding specification of the patterns and synergies.
Figure 4 and 7. What is $\cos \alpha$ as a measure of consistency? The explanation in the Methods does not provide sufficient details to fully understand	At line 215 we now specify the meaning of $\cos \alpha$. Moreover in Methods starting from line 763 we now detail the consistency measures.

Figure 5. What is the difference between patterns consistency, synergies consistency and diagonality?	We introduced new text starting from line 213 by adding: “Activation coefficients of the space-by-time decomposition represent the level of activation of each possible pair of spatial and temporal modules⁴⁵. Consistency was computed as the average scalar product ($\cos\alpha$) of the basic patterns or the synergies across all possible pairs of strides. This scalar product estimates the similarity of a given pattern or synergy across strides. Diagonality was computed as the average ratio of the diagonal activation coefficients to all activation coefficients (Eq. 6 in Methods). We then considered how these consistency measures change as a function of the number of modules (from 1 to 8), by taking the slope of this function (lower panels in Fig. 4b). Maximum slope locates the point of the function where increasing or decreasing the number of modules relative to this point leads to the most drastic change of the corresponding consistency parameter. Thus, the slope represents a sensitive indicator for the dimensionality of modules in the presence of noise.”
Why are there so many individuals with only 1 module even in preschoolers? What does this mean?	The reviewer was right in picking up a number of participants with 1 module. We looked again at the original data (all data, not just those yielding 1 module) and found that indeed in a certain (very low) percentage of cases the maximum of the slope of consistency was poorly defined. We reconsidered these cases and based the new identification of the number of modules on both the consistency measures across strides and the VAF. This is explained in detail in Methods (lines 767 and following). As a result, the number of cases with only 1 module is smaller than before in the revised Fig. 5c. Nevertheless, there is still a not negligible number of legitimate cases with 1 module in all age groups, except the adults. To take the reviewer’s comment into account, at line 254 we added the specification “In particular, one can notice in Fig. 5c that a few participants of all age groups, except the adults, showed only 1 module of neuromuscular activity. This was due to low-frequency oscillations of ipsilateral EMG activities that predominated on all other components.”

REVIEWERS' COMMENTS:

Reviewer #1 (Remarks to the Author):

I appreciate the authors' efforts. I am satisfied with the revised manuscript.

Reviewer #2 (Remarks to the Author):

The authors have revised the manuscript very well and have answered all of my questions in the previous review. It is reassuring that in the newly provided simulations, the slope of the consistency measures can still provide a correct estimation of the data dimensionality when the underlying basic patterns are corrupted by noise. This method of estimating the number of muscle synergies will be a valuable addition to the literature.

I have no further comment and would be glad to recommend the publication of this MS in Communications Biology.

-Vincent CK Cheung

Reviewer #3 (Remarks to the Author):

The authors have addressed my comments well. While I leave it to the other reviewers to comment about how their comments were addressed, I find overall that the revised manuscript is improved and commend the authors for their efforts.

The only minor comment i have left is that it seems that the % sign was used to denote the range of confidence intervals and i wonder if it could lead to confusion as to whether it is a quotient or a range.

We thank again the Reviewers for their evaluation of our study and comments/suggestions that served to improve the manuscript

Reviewer #3	Authors' response
The only minor comment I have left is that it seems that the % sign was used to denote the range of confidence intervals and I wonder if it could lead to confusion as to whether it is a quotient or a range.	We replaced the ÷ sign and used square brackets to denote the range of confidence intervals